



# The evolution of AMULSE (Atmospheric Measurements by Ultra-Light Spectrometer) and its interest in atmospheric applications. Results of the Atmospheric Profiles Of GreenhousE gasEs (APOGEE) weather balloon release campaign for satellite retrieval validation

Lilian Joly[1], Olivier Coopmann[2], Vincent Guidard[2], Thomas Decarpenterie[1], Nicolas Dumelié[1], Julien Cousin[1], Jéremie Burgalat[1], Nicolas Chauvin[1], Grégory Albora[1], Rabih Maamary[1], Zineb Miftah El Khair[1], Diane Tzanos[3], Joël Barrié[3], Éric Moulin[3], Patrick Aressy[3], and Anne Belleudy[3]

[1]GSMA, UMR CNRS 7331, Université de Reims, U.F.R. Sciences Exactes et Naturelles, Reims, France
[2]CNRM, Université de Toulouse, Météo-France, CNRS, Toulouse, France (NWP team)
[3]CNRM, Université de Toulouse, Météo-France, CNRS, Toulouse, France (Instrumentation team)

**Correspondence:** L. Joly, GSMA, UMR CNRS 7331, Université de Reims, U.F.R. Sciences Exactes et Naturelles, Reims, France (lilian.joly@univ-reims.fr); O. Coopmann, Météo-France, CNRM/GMAP/OBS, 42 Avenue Gaspard Coriolis, 31057 Toulouse Cedex, France (olivier.coopmann@umr-cnrm.fr);

**Abstract.**

We report in this paper the development of an embedded ultralight spectrometer (< 3 kg) based on tuneable diode laser absorption spectroscopy (with a sampling rate of 24 Hz) in the mid-infrared spectral region. This instrument is dedicated to in-situ measurements of the vertical profile concentrations of three main greenhouse gases: carbon dioxide ($CO_2$), methane

5   ($CH_4$) and water vapour ($H_2O$) under weather and tethered balloons. The plug and play instrument is compact, robust and cost-effective, autonomous, having a low power consumption, a non-intrusive probe.

It was first calibrated during an *in situ* campaign on an ICOS (Integrated Carbon Observation System) site for several days, then used in a tethered balloon campaign and for a balloon campaign with several balloon flights up to 30 km altitude in the Reims-France in 2017-2018 in collaboration with Météo-France/CNRM.

10   This paper shows the valuable interest of the data measured by AMULSE instrument during the APOGEE measurement campaign, specifically for the vertical profiles of $CO_2$ and $CH_4$, which remain very sparse. We have carried out several experiments showing that the measured profiles have several applications: for the validation of simulations of infrared satellite observations, for evaluating the quality of chemical profiles from Chemistry Transport Models (CTM) and for evaluating the quality of retrieved chemical profiles from the assimilation of infrared satellite observations. The results show that the sim-

15   ulations of infrared satellite observations from IASI and CrIS instruments performed in operational mode for NWP by the Radiative Transfer Model (RTM) RTTOV are of good quality. We also show that the MOCAGE and CAMS CTMs modeled ozone profiles fairly accurately and that the CAMS CTM represents the methane in the troposphere well compared to





MOCAGE. Finally, the measured *in situ* ozone profiles allowed us to show the good quality of the retrieved ozone profiles by assimilating ozone-sensitive infrared spectral radiances from IASI and CrIS.

Keywords. greenhouse gases; atmosphere; ultralight; spectrometer; balloons; in-situ; vertical; Radiative Transfer Model; Numerical Weather Prediction, Chemistry Transport Model, IASI, CrIS, CAMS, MOCAGE

# 1   Introduction

The climate of the Earth currently undergoes a quick change. During last decades, the evidence that this climate change is directly related to the human activities are accumulated (Stocker et al., 2013). The abrupt acceleration of technological progress, the explosion of the industrial activities and agricultural, as well as the multiplication of the means of transport, involve a pro-
found change of our environment, gradually modifying the chemical composition of the atmosphere at a global level on Earth. In particular, the minority chemical compounds in the atmosphere have a fundamental impact on the regulation of the radiative balance of the planet. Indeed, certain gases have the property to absorb a part of the terrestrial infrared radiation. These gases called "GreenHouse Gas" (GHG) are naturally present in the atmosphere (water vapor ($H_2O$), carbon dioxide ($CO_2$), methane ($CH_4$), nitrogen oxide ($NO_x$), ... ). The energy that they collect is then returned in all the directions, at the same time towards
space, but also towards the surface and the various layers of the atmosphere. It is the natural greenhouse effect which makes it possible to have average surface temperature of approximately 15°C instead of -18°C if the atmosphere were transparent with the terrestrial radiation. Any modification of the atmospheric concentration of a GHG induces a modification of the climatic equilibrium. Thus, the atmosphere is certainly the medium most rapidly affected by the disturbances of the equilibrium of the environment, whether natural, as during major climate cycles or linked to human activities. The main greenhouse gases related
to the human activities are $CO_2$, the $CH_4$ and $N_2O$.

The satellite observations have brought many informations of atmospheric composition using the hyperspectral infrared sounders such as IASI (Infrared Atmospheric Sounding Interferometer) or CrIS (Cross-Track Infrared Sounder). These observations are crucial to the study, understanding and follow-up of the atmospheric compounds to monitor the greenhouse gases.
Météo-France operational Numerical Weather Prediction (NWP) systems use RTTOV as a Radiative Transfer Model (RTM) during data assimilation, as many other NWP centers. In order to assimilate the satellite sounder observations, the actual observations have to be compared to the simulation from the model state with a RTM. To accurately simulate infrared observations, RTTOV uses chemical reference profiles that are constant in time and in space. This approximation may lead to possible errors in the simulations. The quality of simulations is essential since the information extracted from these models is then used in
data assimilation systems for weather forecasting. Which is why there is a need to assess the quality of chemical profiles with



*in situ* measurements.

In addition to the NWP models, Chemistry Transport Models (CTM) are available. Indeed, air pollution is a public health issue especially in big cities and would be responsible for 790,000 deaths per year just in Europe (Lelieveld et al., 2019). One

of the main pollutants in the troposphere is ozone which increases the death rate during pollution episodes. At large scales, the effect of the global change continues to be important due to the increase of GHG whose main contributors are carbon dioxide and methane. CTM such as MOCAGE (MOdèle de Chimie Atmosphérique à Grande Échelle) at Météo-France and C-IFS (atmospheric Chemistry in the Integrated Forecasting System) of CAMS (Copernicus Atmospheric Monitoring Service) at ECMWF (European Centre for Medium-Range Weather Forecasts) allow to produce high-quality forecast of chemistry field

in the stratosphere and upper troposphere but forecast quality is weaker in the UTLS (Upper Troposphere Lower Stratosphere). However, satellite sounding in the atmospheric boundary layer is more difficult, especially for infrared sensors because of cloud and aerosols that interfere with the signal; land surface emissivity and temperature uncertainties also are part of the problem. The data assimilation of satellite observations sensitive to atmospheric composition allows to obtain accurate chemical description of the atmosphere especially in the UTLS. Presently, C-IFS assimilates Level 2 products from several satellite

instruments. To assess the quality of the CTM forecasts of chemical fields, we need to have accurate measurements of these compounds on the atmospheric column. Despite many chemistry measurements at ground stations, informations on chemistry is not widely available at high altitudes. To overcome this lack of data, many projects have been started such as the APOGEE (Atmospheric Profiles Of GreenhousE gasEs) campaign. APOGEE campaign derives from a collaboration between GSMA (Groupe de Spectrométrie Moléculaire et Atmosphérique) at Reims Universiy in France, LSCE (Laboratoire des Sciences du

Climat et de l'Environnement) in France and CNRM (Centre National de Recherches Météorologiques) at Météo-France. The objective of APOGEE campaign is to realize measurements of pressure, temperature, humidity and main atmospheric chemical vertical profiles ($CO_2$, $CH_4$, $O_3$ and $H_2O$) up to 30 km included atmospheric boundary layer. Ozone profile is measured using Vaisala radiosondes with electrochemical cell. $CO_2$, $CH_4$ and $H_2O$ are measured using AMULSE (Atmospheric Measurement Ultalight SpEctrometer) instrument developed by GSMA.

In Section 2, we will identify some ways to measure the chemical composition of the atmosphere. Then in Section 3, we will describe the different characteristics of the AMULSE instrument as well as comparisons with another instrument. Finally, we will use the data from the APOGEE measurement campaign on one study case to evaluate the sensitivity of infrared observation simulations on $CO_2$, $CH_4$ and $O_3$ information and we will use the *in situ* profiles to evaluate *a priori* profiles from the CTMs

and retrieval profiles by 1D assimilation experiments.



## 2 Atmospheric composition measurements

### 2.1 Molecules of interest

$CO_2$ and $CH_4$ are long-lived gases. Atmospheric residence time is a few decades for $CH_4$ whose reactivity makes it an important player in atmospheric chemistry (Voulgarakis et al., 2013) and a few hundred to thousands of years for the $CO_2$ that is inert to

it in the atmosphere from a chemical point of view (Archer and Brovkin, 2008; Eby et al., 2009). The increase of these gases in the atmosphere is conditioned by their anthropogenic emissions which add up to an active natural cycle. The $CO_2$ cycle is the carbon cycle as this gas dominates the atmospheric composition of carbon compounds in terms of mass (nearly 215 times higher than that of methane). However, the effectiveness of $CH_4$ in absorbing infrared radiation is much greater than that of carbon dioxide. Since the beginning of the industrial era in 1750, the mixing ratio of atmospheric $CO_2$ has increased from around

280 ppm to 401 ppm till October 2016, an increase of more than 40%. The increases in the $CO_2$ and $CH_4$ levels along with the uncertainty of the $H_2O$ at high altitudes upset the radiative balance of the planet. Therefore, having information and data about the vertical distribution of these three GHG is very useful to improve our knowledge of the future of our climate. Hence, we should improve the knowledge and estimation of the regional anthropogenic GHG natural sinks and emission sources for a better quantification : 1) by enhancing the atmospheric chemistry-transport models that are used to link the sources and the

sinks to the atmospheric concentrations; 2) by increasing the atmospheric observation and measurements.

### 2.2 GHG atmospheric observations

The goal of atmospheric observations of GHG is to follow the evolution of these gases. According to the IPCC (Intergovernmental Panel on Climate Change), it is necessary to be able to determine both the long-term trend related to global emissions, a seasonal cycle linked to vegetation activity or the availability of OH, and a synoptic variability linked to the transport of

air masses over periods of time from few hours to few days. Typically, in one year, the air of the Northern Hemisphere is mixed with that of the Southern Hemisphere and vice versa. At mid-latitudes in the northern hemisphere, emissions are transported around the Earth in a few days by a zonal circulation of air masses that is much more efficient than mixing at latitudes. By nature, in-situ measurements, therefore, require continuous and diversified observation means on a global scale: ground measurements (ICOS , WMO , ...), airborne measurements (CAMMAS et al., 2008; Filges et al., 2015; Nédélec et al., 2015;

Petzold et al., 2015), satellite observations (Crevoisier et al., 2013; Thompson et al., 2012; Wecht et al., 2014) and vertical measurements using balloons (Ghysels et al., 2016). Tethered balloons measurements can be used up to 800 meters' altitude and they are cheap and allowed in France. They do not require lots of preparations and logistics while they offer controlled travelling speed as well as fixed point measurements along with good payload options. On the other hand, weather balloon offers high altitude measurements (up to 30 km) but the fixed point measurement cannot be realized. In our previous papers

(Joly et al., 2016; Khair et al., 2017) we discussed the advantages and disadvantages of the different ground and airborne measurement techniques as well as the need for observations on the vertical along the atmospheric column in order to complete the spatial measurements. To complement the current observational system, stratospheric balloons are unique scientific research tools for accessing the stratosphere, an area inaccessible to airborne measurements. The information provided by the satellites





is an average integrated on all or part of the atmospheric column except for a few measurements taken at the limb that provide information with a low vertical resolution. This specificity of the balloons, to be able to access the pofiles, makes it an ideal tool to explore the distribution of numerous atmospheric parameters between the surface and 40 km of altitude in a strategic zone where the masses of air mix a large number of particles and chemical compounds emitted from the Earth. We report in

this paper the development of a lightweight instrument called AMULSE (for Atmospheric Measurements by UltraLight SpEctrometer) that fulfills the requirements for weather balloons flights in order to increase the atmospheric GHG measurements. This instrument was first tested at ground level during an intercomparison with PICARRO's instrument of the LSCE laboratory in Paris member in the ICOS network. It was then deployed into *in situ* measurements on tethered balloons and on weather balloon up to the stratosphere at about 30 km altitude for $CO_2$ and $CH_4$ while we measured simultaneously the $H_2O$ up to 10

km altitude in order to calculate the mixing ratios in dry air column. High precision measurements and vertical resolution of few meters in-situ concentration profiles is achieved by using diode-laser spectroscopic technique (Durry and Megie, 1999a; Joly et al., 2007, 2016) combined with weather balloon and tethered balloons. The developed optical sensor is then based on mid-infrared absorption spectroscopy which provides a compact, cost-effective, fully autonomous, low-power consumption and non-intrusive probe to measure the targeted gases in the atmosphere using an open-path multipass cell.

## 3 AMULSE

In recent years, the AMULSE instrument has evolved in order to offer a single-gas $CO_2$ version in 2014, a single-gas $CH_4$ version in 2015, a dual-gases version in 2016 ($CO_2$/$CH_4$) and a tri-gases version ($CO_2$/$CH_4$/$H_2O$) in 2017, still weighing less than 3 kg. We are also working to improve the accuracy and robustness of the instrument.

### 3.1 Principle

The high selectivity and sensitivity in the gases detection realized using the diode laser absorption spectroscopy is considered as the most advantageous technique for atmospheric sensing (Durry and Megie, 1999b). It is based on semiconductor diode lasers because they offer a continuous mode emission, they are tunable and have a relatively low noise amplitude. The diodes used in this work emit in the near infrared spectral region (NIR) where most atmospheric pollutant molecules feature suitable absorption lines. Direct Absorption Spectroscopy is the simplest application of this technique and it is well adapted to in-situ

measurements. It requires that the tunable laser beam with an intensity of $I_0$ passes through the gas sample on a distance L and then measured using a detector. When the frequency of the emitted light is close to a molecular transition $\nu_0$ of the gaseous sample, the light is then absorbed and the transmitted intensity $I(\nu)$ decreases. The concentration of the absorbing species in the gas mixture is then calculated according to Lambert Beer's law.

### 3.2 Technical description

Its architecture is similar to that used in our previous papers (Joly et al., 2016; El Khair et al., 2017).This new version is equipped with two lasers (purchased from nanoplus GmbH, Gerbrunn, Germany). The first one is a GaSb-based DFB semi-





conductor diode emitting at 2.004 $\mu m$ . A simple change on its scanning ramp allows us to target the $CO_2$ (4992.51 cm-1) and $H_2O$ (4992.94 cm-1) molecules simultaneously on the same scanning window. The second laser is an ICL DFB diode emitting at 3.24 $\mu m$ detailed by (El Khair et al., 2017) used to target the $CH_4$ molecules (R(6) transition at 3085.86 cm-1). The lasers driving current is ramped at 24 Hz (42 ms) ensuring a good spatial resolution (1-20 m) depending on the ascent and decent

flight speed. After collimation, the laser beams pass then into a home-made open-path Herriott multipass cell achieving an optical path-length of 16 m on the $CO_2$/$H_2O$ channel and 18 m on the $CH_4$ channel. The mirrors are heated in order to avoid the condensation on their surfaces using 2 W heaters.

At the output of the cell, both laser beams are focalized onto the two photodetectors (purchased from Judson, Montgomeryville, PA, USA). The central processing unit (National Instrument real time) records all data (spectra, PTU and GPS

from the on-board meteorological radiosonde. The spectrometer weights is lower than 3 kg in flight-ready condition and power supplies ensuring about 6h of operation. When operated under weather balloons, the spectrometer is equipped with a satellite communication system based on an Iridium element. The Iridium module send various datas such as meteorological fields and GPS localisation as well as monitoring parameters of the instrument during the flights. Under a tethered balloon, a Wi-Fi module replaces the Iridium, allowing to send to a computer fixed at 400 m the recorded spectra in order to achieve nearly

real-time data processing and hence determine the position and the concentration of the plume.

### 3.3   Amulse compared to Picarro (2015-2018)

Each time AMUSLE evolved, we made different comparisons with measurements from an ICOS site in order to characterize the impact of instrumental modifications. The table shows the improvement in sensitivity between 2015 and 2018 (Table 1). This improvement comes from the optimization of the optical cell, electronics and spectroscopy.

|         | 2015    | 2017    | 2018    |
|---------|---------|---------|---------|
| $CH_4$  | 97 ppb  | 14 ppb  | 2.7 ppb |
| $CO_2$  | 1.4 ppm | 2.6 ppm | 0.5 ppm |
| $H_2O$  |         | 50 ppm  | 18 ppm  |

**Table 1.** This table illustrates the evolution (bewteen 2015 and 2018) of AMULSE sensitivities with respect to years and measurable molecular species. It highlights an increasing improvement in the performance of the instrument.

In order to better understand the values in this table, we will describe the last comparison measurement campaign that acheived in February 2018. It took place at the LSCE on their ICOS site where a Picarro's instruments (model G1301, Picarro Inc., Santa Clara, CA, USA) is installed. All Picarro's concentration are calibrated (every 6 hours) with a WMO (World Meteorological Organization) standard gas. The analyser pulled air continuously at few centimeters from the AMULSE about more than three continuous days. Figure 1 shows the intercomparison between the two instruments with Picarro's data plotted

in green and Amulse's data in blue. Figure 1 (a),(b) and (c) correspond to a zoom on 10 minutes. The results show a good correlation between the two instruments for the three measured species. The mean difference between the two instruments for





the $CO_2$ channel is equal to 0.01 ppm with a standard deviation of 0.5 ppm, 0.1 ppb with a standard deviation of 2.7 ppb for the $CH_4$ channel and 2 ppm with a standard deviation of 18 ppm for the $H_2O$ channel

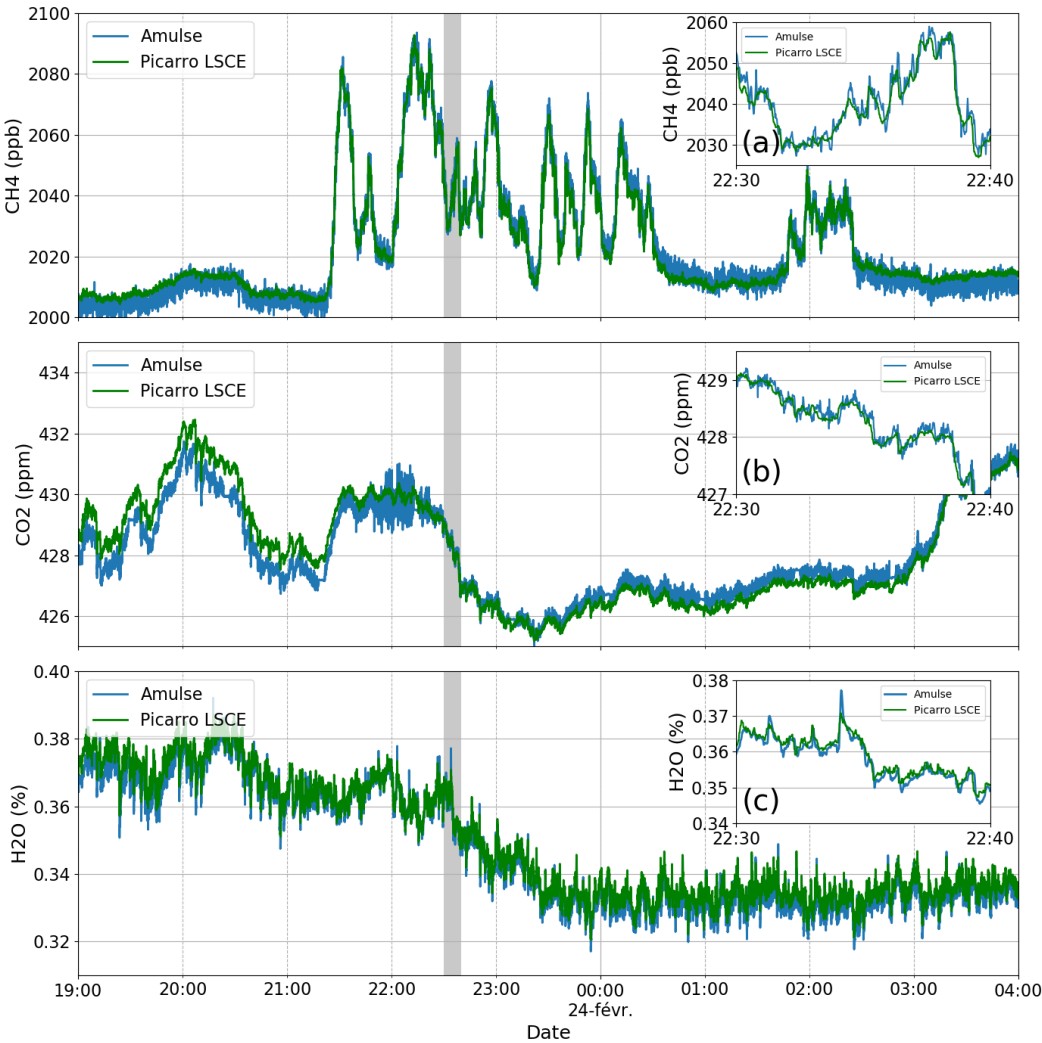

**Figure 1.** Intercomparsion between Amulse's concentration (in blue) and Picarro's concentration calibrated with WMO gases (in green) during 9 hours for $CO_2$, $CH_4$ and $H_2O$. Gray areas correspond to the zoomed views of (a) $CO_2$,(b) $CH_4$ and (c) $H_2O$

## 4 Atmospheric applications

Once the validation of the instrument was realized with the help of the PICARRO's, several atmospheric measurement cam-
5   paigns were carried out.



## 4.1 Tethered balloon application

Tethered balloons offer great opportunities in order to characterize the temporal atmospheric evolution of the three measured species up to 800 m altitude. In such application we can control the motion speed of the balloon, we can have a bigger payload (which depends on the type of the balloon used), we can even acquire data from a fixed stationary point. Tethered balloon is

5    a carrier that allows measurements to be made between a fixed tower and under-aircraft measurements. The costs of a captive balloon are much lower than the installations mentioned above.

A campaign was conducted to monitor the evolution of the atmospheric boundary layer early in the morning at sunrise. We carried out 21 ascents/descents (up to 50 metres above sea level). The time of an ascent is about 6 minutes, which results in a spatial resolution of between 20 and 50 cm.

10    These results were first interpolated over all the heights so that we visualized the spatio-temporal evolution of the boundary layer that morning (Figure 2). AMULSE simultaneously records the concentration of $CO_2$ (top left), $CH_4$ (top right), $H_2O$ (bottom left) and temperature (bottom right).

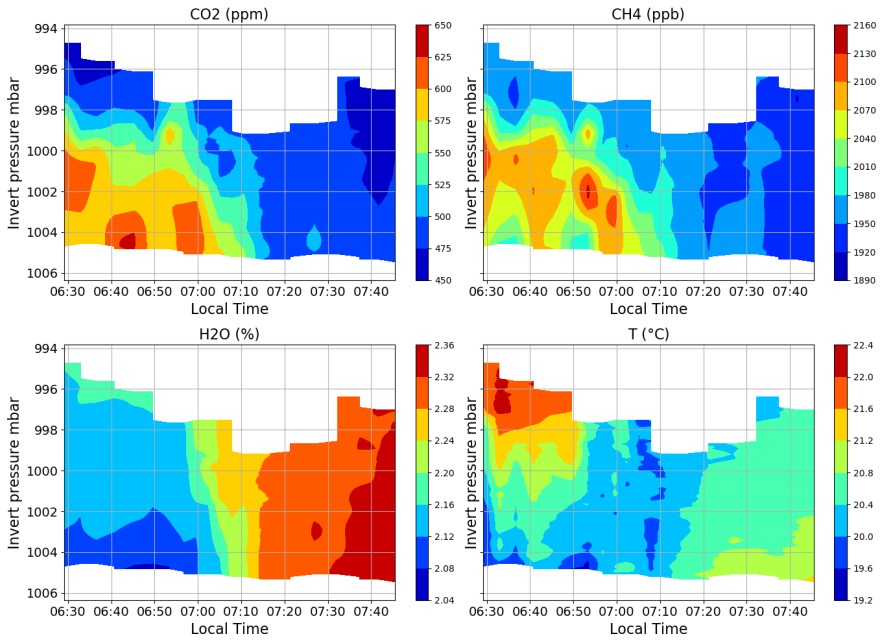

**Figure 2.** Atmospheric boundary layer evolution. Here is an example of the results of the AMULSE measurements under captive balloon, it allows to measure simultaneously as a function of time and altitude, the concentration of $CO_2$ (top left), $CH_4$ (top right), $H_2O$ (bottom left) and temperature (bottom right)

This illustrates the interest of this instrument on board a captive balloon, as it allows the study of $CO_2$, $CH_4$ and $H_2O$ of the boundary layer.



## 4.2 APOGEE Campaign

### 4.2.1 Objectives

One of the goals is to carry out weather balloon measurements in co-location with the IASI satellite, in orbit around the earth, developed jointly by CNES (Centre National d'Etude Spatiale) and EUMETSAT (European Organisation for the Exploitation

of Meteorological Satellites). The measurement made by AMULSE instrument will provide comparative data that can be used to validate CNRM/Météo France meteorological models. In addition, the combination of all these measures offers an opportunity to analyse physico-chemical processes that have not yet been studied in the stratosphere and the interface between the troposphere and the stratosphere, commonly known as UTLS (Upper Troposphere - Lower Stratosphere). To achieve these objectives, we carried out measurements from the GSMA site (49°14'29.608" N, 4°4'4.709" E) using meteorological balloon

for the measurements of $CO_2$, $CH_4$, $H_2O$, $O_3$, P, T, U in co-location with Metop A, Metop B and Suomi-NPP

To measure all these parameters, the instruments used were as follows:

- Vaisala RS92-SGP radiosondes (pressure, temperature, relative humidity and GPS location measurement every second with a real time transmission)

- Electrochemical concentration cell (ECC) ozonesondes from Science Pump Corporation, models 5A and 6A with Vaisala

RS-92 interfacing (ozone concentration measurement every second with a real time transmission)

- The AMULSE instrument presented above

### 4.2.2 Description of the flight chain

Flight chain is composed of a carrier balloon, a parachute and the payload (AMULSE and radiosonde) (Figure 3a). Balloon is inflated using a tare to ensure good repeatability of ascent speed. A wire is connected to both parachute and AMULSE release

system; the carrier ballon is hooked up to the wire using a sliding ring. Balloon is released when the wire is cutted and it can be triggered either at a specific atmospheric pressure (fixed before the launch), or via a satellite communication system based on an Iridium modem embedded on the AMULSE electronic board or at a specific timeout fixed in advance. When the carrier balloon is released, the wire slides out of the ring and the instrumentation descends with the parachute at a speed lower than 5m/s (Figure 3b). A meteorological radiosonde with a GPS probe fixed on the instrumentation and connected to

a ground station allows the determination of the landing site which helps recovery of the instrumentation. A flight simulation software is used to estimate the trajectories and the landing point of the probe. This estimation is based on the wind forecasts from both Météo-France operational numerical weather prediction models: Applications de la Recherche à l'Opérationnel à Méso-Echelle (AROME) (Seity et al., 2011; Brousseau et al., 2016) and Action de Recherche Petite Echelle Grande Echelle (ARPEGE) (Courtier and Geleyn, 1988; Déqué et al., 1994). Hypothesis are made on both ascending/descending speed of

the system and the release of the carrier balloon in order to run a first simulation. A measured vertical profile of horizontal winds by the recent sounding can also be used. The trajectories of the balloon are updated in real time by the rawinsonde RS92





measured wind. As the trajectories are adjusted in real-time and AMULSE release system can be remotely triggered, landing area can be optimized in order to maximize safety landings.

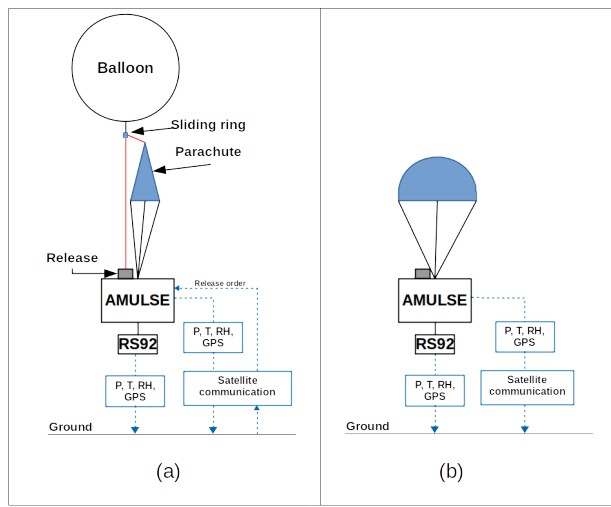

**Figure 3.** A schematic diagram of the flight chain during (a) ascent and (b) descent. Transmission and reception of the information during the flight is done using both satellite and radio communications. A Vaisala radiosonde RS92 is connected in order to send instantaneous P, T, relative humidity and GPS data to a ground mobile station to inject those data in the trajectory model and hence ensure the tracking/recovery of the instrument. Note that the trigger release system can be controlled by either home-made smartphone application or web application using Iridium satellite communication system. Theses applications can also track AMULSE position as the latter transmits its GPS position, pressure and temperature data every 5 min ensuring redundant retrieval system.

### 4.2.3   Performed Flights

The measurement campaign was held in France in the Champagne - Ardenne region. The launching site was located at the
5   Campus Moulin de la Housse, in Reims. The flights took place over the period from November 2017 to April 2018 (Table 2) and was carried out either by day or night. Several launches could be done at the same time (Figure 4).

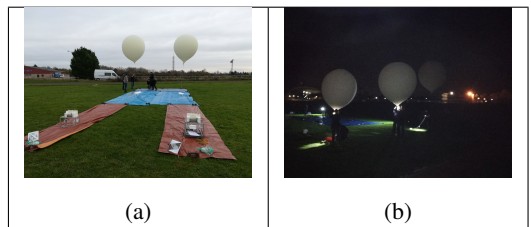

**Figure 4.** Launching site (a) at day and (b) at night, ready for multiple launches.





| Flight | Date | Max altitude (m) |
|--------|------|------------------|
| 1 | 07/11/2017 13:48 UTC | 19121 |
| 2 | 08/11/2017 11:50 UTC | 25460 |
| 3 | 15/11/2017 12:02 UTC | 31156 |
| 4 | 15/11/2017 12:09 UTC | 31388 |
| 5 | 15/11/2017 12:06 UTC | 31110 |
| 6 | 13/04/2018 12:06 UTC | 29490 |
| 7 | 17/04/2018 10:15 UTC | 25000 |
| 8 | 17/04/2018 20:12 UTC | 19660 |

**Table 2.** List of flights between November 2017 and April 2018

Figure 5 shows examples of vertical concentrations profiles of $CO_2$, $CH_4$, $H_2O$ obtained from the flights. The concentration of CO2 on the ground can vary greatly, depending on the time of measurement because the boundary layer is enriched with $CO_2$ when flying close to night (photosynthesis phenomenon). Concerning $CH_4$, we see that there is a slight difference but the decrease always occurs at the tropopause level. Its decrease is due to its oxidation with the OH radical. For water measure-

ments, we are currently using the "Imet 1" radiosonde. The measurements are consistent below 10 km altitude but above this altitude we consider that the measurements are biased. For this reason, one of the prospects for 2020 is to make water vapour measurements by laser diode spectrometry to have a better accuracy in the stratosphere. It should also be taken into account that we generally see an increase in water vapour concentration above 10 km, this is due to the degassing of water from the instrument and the balloon during the ascent. For this reason, water measurements in the stratosphere are only possible during

descents.

## 5   Use of data from the APOGEE measurement campaign

The data produced during the APOGEE measurement campaign have different applications. Initially, they were used as satellite validation data for the IASI and CrIS infrared sounders. They were then used to evaluate the quality of vertical ozone and methane profiles extracted from the MOCAGE and CAMS Chemistry Transport Models. Finally, they were used as verification

data to compare the ozone profiles returned in one-dimensional variational assimilation experiments (1D-Var) with the *a priori* ozone profiles from MOCAGE and CAMS.

### 5.1   Sensitivity of infrared satellite sensors to $CO_2$, $CH_4$ and $O_3$ informations

#### 5.1.1   IASI and CrIS sensors

IASI is flying on board 3 European polar-orbiting satellites Metop-A, B and C, respectively launched in 2006, 2012 and

2018. The IASI spectrum covers the range between 645 and 2760 $cm^{-1}$, with as spectral sampling of 0.25 $cm^{-1}$ and spectral





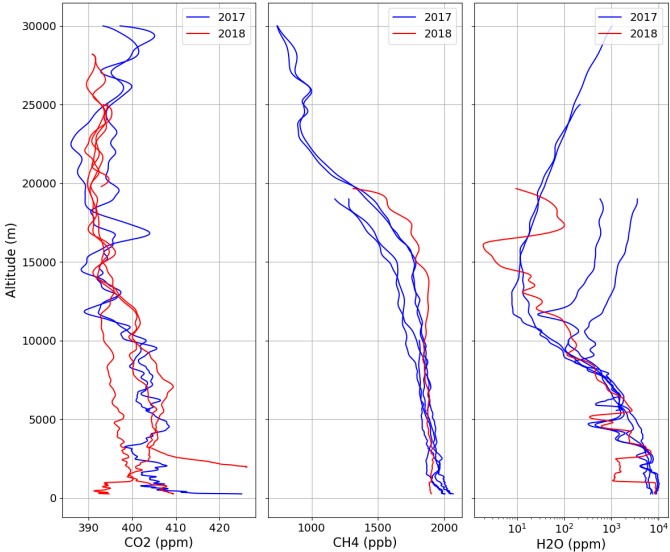

**Figure 5.** Balloon measurement : $CO_2$, $CH_4$, $H_2O$ vertical concentration profiles between November 2017 and April 2018

resolution of 0.5 cm$^{-1}$ after apodisation. CrIS is part of the payload of the US polar-orbiting satellites S-NPP and NOAA-20, respectively launched in 2011 and 2017. Only the so-called CrIS Normal Spectral Resolution data are used in this study, leaving the CrIS spectrum divided into three parts: between 650 and 1095 cm$^{-1}$ with a spectral resolution of 0.625 cm$^{-1}$, between 1210 and 1750$^{-1}$ (1.25 cm$^{-1}$ spectral resolution) and between 2155 and 2550 cm$^{-1}$ (2.5 cm$^{-1}$ spectral resolution).

### 5.1.2 Radiative Tranfer Model

During the assimilation process within the NWP analysis step, a compromise is found between a prior knowledge of the atmospheric state (a short-range forecast, in general) and various observations, including radiosoudings, aircraft measurements, ground stations, space-borne passive sounders both in the microwave and in the infrared, among others. When assimilating satellite radiances, eg. infrared radiances, a radiative transfer model is needed in several ways. Firstly, the observations have to be compared to the prior atmospheric state: simulations of radiances are made using the temperature and humidity profiles and surface parameters from that state. The differences between observations and simulations are called innovations. Then, the information contained in the innovations is used to modify the atmospheric state so that the analyzed atmospheric state is a compromise between all sources of information. To achieve this inverse problem, the RTM has to either have tangent linear and adjoint codes or compute the jacobians (the Jacobians of IASI channels are described in Section 5.1.3). At Météo-France, the RTM used in the operational assimilation software is the fast radiative transfer model RTTOV (Radiative Transfer for Tiros Operational Vertical sounder) (Saunders et al., 2018) developed and maintained by the NWP-SAF (Satellite Application Facility) of EUMETSAT (EUropean Organisation for the Exploitation of METeorological SATellites).





For infrared sensor simulation, RTTOV requires not only temperature and humidity profiles, but also a knowledge of the atmospheric composition. As few NWP models have in-line atmospheric composition yet or are coupled to a CTM, RTTOV also provides the users with average chemical profiles that are invariant in time and space, hereafter referenced as REF profiles. These REF profiles are plotted on Figure 7 for $O_3$ (a), $CO_2$ (b) and $CH_4$ (c). In this study, we use RTTOV version 12, with
coefficients for IASI and CrIS on 101 fixed pressure levels.

### 5.1.3 Theoretical sensitivity of IASI and CrIS spectra to GHG

Infrared satellite observations measured by IASI and CrIS instruments are sensitive to atmospheric temperature and humidity, skin temperature but also atmospheric chemistry. To identify the channels sensitive to the different parameters, RTTOV offers the possibility to calculate the jacobians which represent the sensitivity of the brightness temperature to the variation of a
thermodynamic or chemical parameter. Thus we have represented in Figure 6, the jacobians in temperature (a), humidity (b), skin temperature (c), ozone (d), carbon dioxide (e) and methane (f) for the 8461 IASI channels using the reference chemical profiles for our case study according to the 101 RTM levels. It should be noted that temperature jacobians are sensitive over a large part of the spectrum but at different altitudes. Indeed, channels between 650 to 770 cm$^{-1}$ describe a temperature sensitivity from the top of the stratosphere to the surface. There are also high sensitivity to window channels between 790 to
980 cm$^{-1}$ and 1.080 to 1.150 cm$^{-1}$. There is a sensitivity over the entire atmosphere for channels between 1.000 to 1.070 cm$^{-1}$. Then, temperature jacobians are sensitive in the troposphere between 1.210 to 1.650 cm$^{-1}$. Humidity jacobians are surface sensitive for channels between 650 to 770 cm$^{-1}$ and 1.080 to 1.150 cm$^{-1}$ and in the troposphere between 650 to 1.150 cm$^{-1}$. Jacobians of skin temperature have essentially high values for window channels between 790 to 980 cm$^{-1}$ and 1.080 to 1.150 cm$^{-1}$. Ozone jacobians are slightly sensitive between 650 to 770 cm$^{-1}$ and have a sensitivity over the entire atmosphere
between 1.000 to 1.070 cm$^{-1}$. We note that carbon dioxide jacobians have high values over the entire atmospheric column for channels between 650 to 770 cm$^{-1}$. Finally, methane jacobians have high sensitivities for channels between 1.210 to 1.650 cm$^{-1}$.

### 5.1.4 Pre-processing of APOGEE measurements

We selected the sounding from 2018-04-17 at 10 UTC for which ozone, carbon dioxide and methane are measured. The pro-
files are plotted in Figure 7. The ozone profile is available up to 6 hPa (TBC), the carbon dioxide up to 25 hPa (TBC) and the methane up to 65 hPa (TBC), as indicated by the horizontal separation on each plot. There are few differences between the reference and *in situ* profiles for ozone and methane, except around the tropopause. Unlike carbon dioxide, where the difference can be as much as 10 ppmv in the lower troposphere.

In order to assess the impact of using the *in situ* chemical information instead of the reference profiles, we had to interpolate the former on the 101 fixed pressure levels of the latter. Moreover, the measured profiles do not reach the highest fixed pressure level. Thus a polynomial function was used to link the *in situ* profiles to the reference profiles from the levels where the data



**Figure 6.** Jacobians of temperature (a), humidity (b), skin temperature (c), ozone (d), carbon dioxide (e) and methane (f) for 8461 IASI channels w.r.t. 101 RTM levels.

are missing. Shaded areas in Figure 7 represent parts or profiles *in situ* are linked to reference profiles to replace missing data.

Then, modeled atmospheric profiles of temperature, humidity, surface temperature, surface humidity, surface pressure, zonal and meridian wind come from the global model ARPEGE forecasts (every 3 hr), which have been extracted for the same





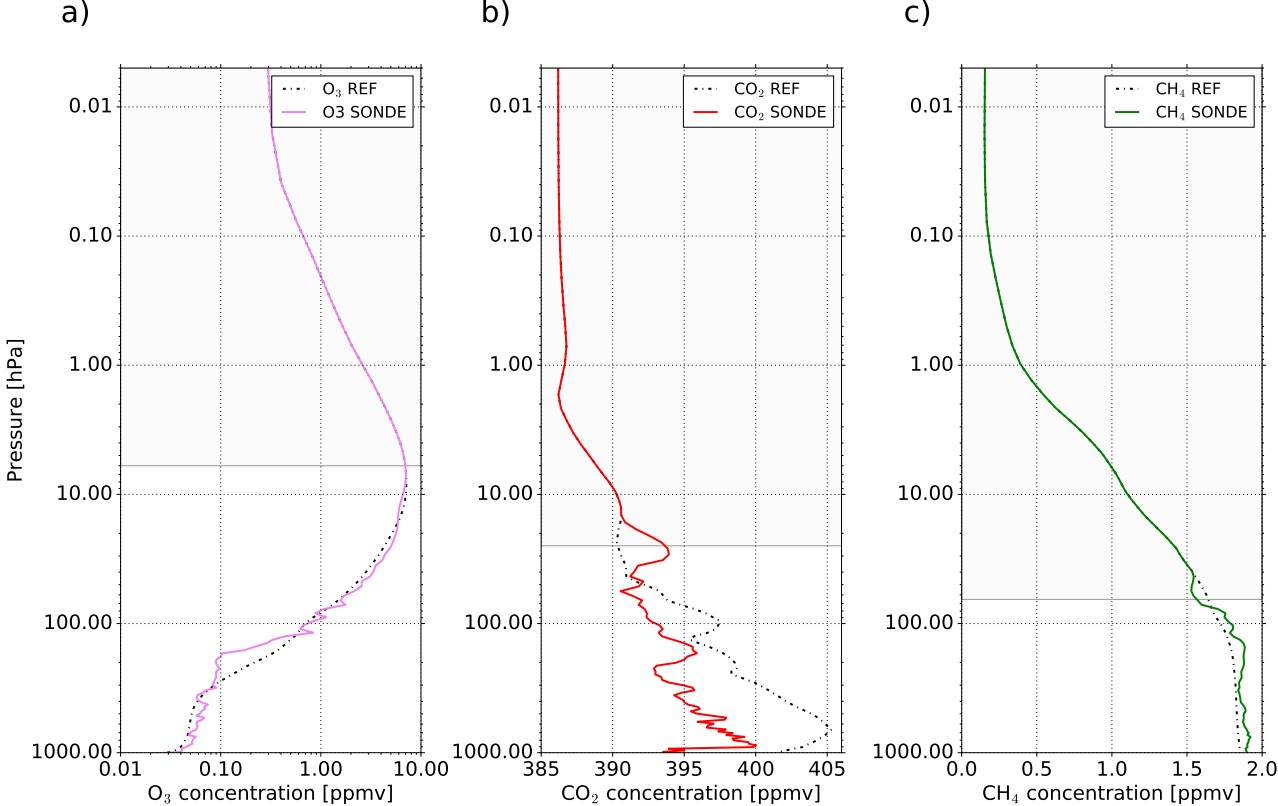

**Figure 7.** Representation of reference profiles (black dotted lines) of $O_3$ (a), $CO_2$ (b) and $CH_4$ (c) and profiles measured by radiosonde and convoluted with reference profiles of $O_3$ in violet (a), $CO_2$ in red (b) and $CH_4$ in green (c). Shaded areas correspond to levels for which there are no measured data.

period and location as for the radiosonde. The coefficient file used during the simulation process in RTTOV being on 101 fixed pressure levels, the thermodynamic profiles of ARPEGE were interpolated on these same levels.

In this study we simulated satellite observations from two types of infrared instruments: IASI and CrIS. In order to evaluate the quality of our simulations, we sought to spatially and temporally co-locate the IASI and CrIS pixels as close as possible
5 to the radiosondes. To avoid problems related to clouds, we carried out the April 17, 2018 release at 10 UTC in clear sky. Thus, Figure 8 represents the brightness temperature spectrum of the IASI observations simulations in black and CrIS in red with respect to the wave number using as RTTOV input the reference profiles of $O_3$, $CO_2$ and $CH_4$ for the radiosonde from 2018-04-17 to 10 UTC. There is a good agreement between the simulated IASI and CrIS spectrum except for the window channels. This is because the IASI and CrIS pixels selected are not strictly in the same place, which implies a different skin
10 temperature used in RTTOV. However, the skin temperature has an obvious impact on the simulation of infrared observations sensitive to the surface, hence these differences.





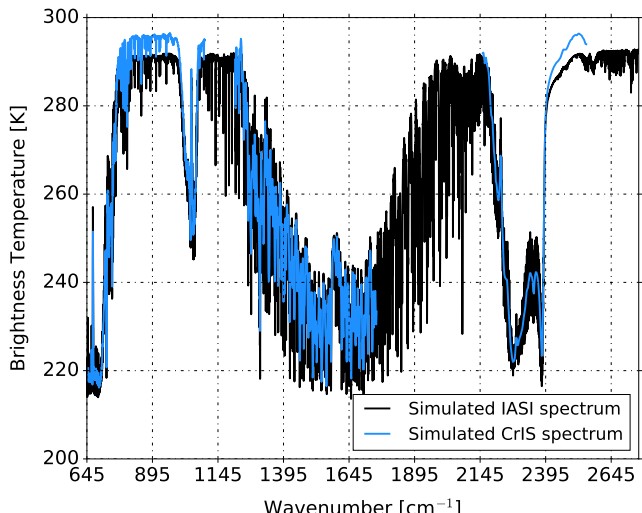

**Figure 8.** Brightness Temperature spectrum of simulated IASI (in black) and CrIS (in blue) observations w.r.t wavenumbers using $O_3$, $CO_2$ and $CH_4$ reference profiles for radiosonde from 2018-04-17 at 10 UTC.

### 5.1.5 Impacts on simulations

We evaluated the impact on the simulations of IASI and CrIS observations of using chemical information from *in situ* measurements as input to RTTOV in replacement of the constant profiles. Figure 9 shows the difference between simulated brightness temperature (BT) with $O_3$ in purple (rp $CO_2$ in red or $CH_4$ in orange) *in situ* profiles from SONDE and AMULSE and simulated brightness temperature with $O_3$ (rp $CO_2$ or $CH_4$) reference profiles for radiosonde from 2018-04-17 at 10 UTC for all channels contained in the IASI (a) and CrIS (b) spectrum, such as:

$$\text{EXPO}_3 = \text{BT}^{\text{simul}}[\mathbf{O_3^{SONDE}} + \text{CO}_2^{\text{REF}} + \text{CH}_4^{\text{REF}}] - \text{BT}^{\text{simul}}[\text{O}_3^{\text{REF}} + \text{CO}_2^{\text{REF}} + \text{CH}_4^{\text{REF}}]$$

$$\text{EXPCO}_2 = \text{BT}^{\text{simul}}[\text{O}_3^{\text{REF}} + \mathbf{CO_2^{AMULSE}} + \text{CH}_4^{\text{REF}}] - \text{BT}^{\text{simul}}[\text{O}_3^{\text{REF}} + \text{CO}_2^{\text{REF}} + \text{CH}_4^{\text{REF}}] \quad (1)$$

$$\text{EXPCH}_4 = \text{BT}^{\text{simul}}[\text{O}_3^{\text{REF}} + \text{CO}_2^{\text{REF}} + \mathbf{CH_4^{AMULSE}}] - \text{BT}^{\text{simul}}[\text{O}_3^{\text{REF}} + \text{CO}_2^{\text{REF}} + \text{CH}_4^{\text{REF}}]$$

The differences of BT affects ozone-sensitive channels located between 710 to 760 cm$^{-1}$, 980 to 1.150 cm$^{-1}$ and 2.060 to 2.135 cm$^{-1}$ (for IASI). Then, the differences of BT affects a larger part of the spectrum especially the $CO_2$ sensitive channels with spectral intervals between: 645 to 820 cm$^{-1}$, 930 to 980 cm$^{-1}$, 1.030 to 1.085 cm$^{-1}$, 1.900 to 1.940 cm$^{-1}$ (for IASI), 2.010 to 2.120 cm$^{-1}$ (for IASI) and 2.200 to 2.440 cm$^{-1}$. Finally, the differences of BT affects $CH_4$ sensitive channels located between 1.200 to 1.380 cm$^{-1}$ and less so between 2.650 to 2.760 cm$^{-1}$.

The maximum difference values of BT for $CO_2$ sensitive channels are around 0.2 K and for $CH_4$ sensitive channels around -0.4 K for both instruments. The maximum difference values of BT for ozone sensitive channels are around 0.9 K for IASI and





0.75 K for CrIS. The differences in $\Delta BT$ values between the two instruments can be explained by the difference in spectral resolution.

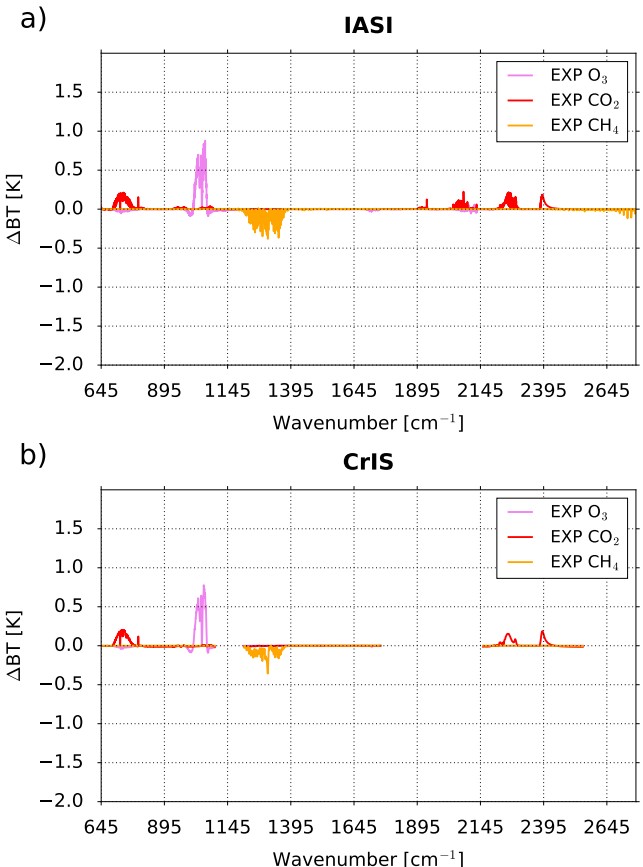

**Figure 9.** Difference between simulated brightness temperature (BT) with $O_3$ in purple (,rp $CO_2$ in red or $CH_4$ in orange) *in situ* profiles from SONDE and AMULSE and simulated brightness temperature with $O_3$ (,rp $CO_2$ or $CH_4$) reference profiles for radiosonde from 2018-04-17 at 10 UTC for all channels contained in the IASI (a) and CrIS (b) spectrum.

Even if the reference profile is close to the *in situ* profile, as for example for ozone, this does not necessarily imply small differences between the simulations. However, the spatial and temporal variability of atmospheric composition is very important, especially for ozone. In addition, differences in BT too large in $CO_2$ have a direct impact on the quality of temperature forecasts, since we use channels sensitive to this species to retrieved temperature profiles.

Finally, in order to assess the quality of our simulations of infrared satellite observations for IASI and CrIS instruments using the reference profiles provided by RTTOV, we calculated the difference between real and simulated observations (O-B) using chemical information from reference profiles (black line) and from *in situ* profiles (red line) with respect to IASI (Figure 10.a)



and CrIS (10.b) monitored channels for radiosonde from 2018-04-17 at 10 UTC. IASI and CrIS has respectively 314 and 330 monitored channels. For this case study the differences (O-B) using reference chemical profiles (REF) and simulations using

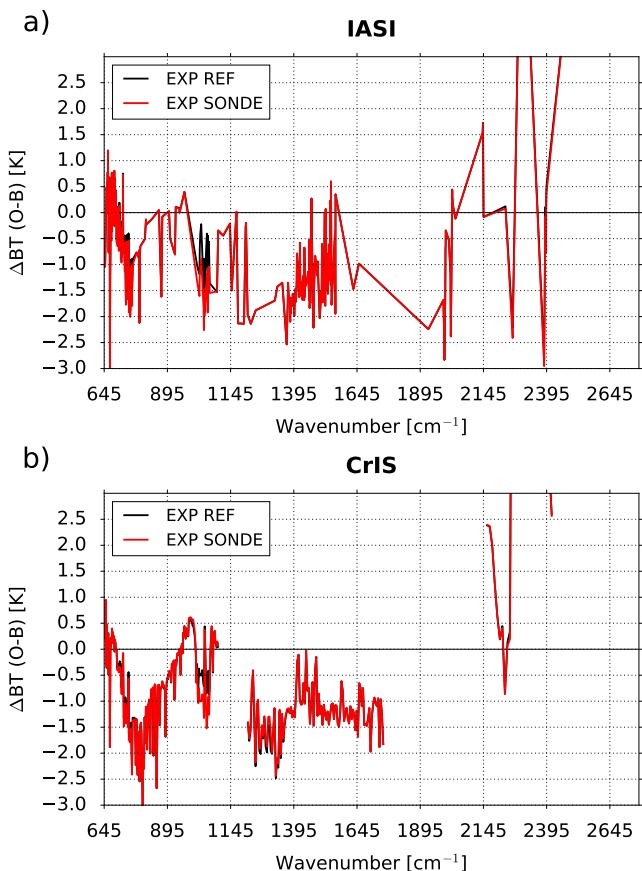

**Figure 10.** Difference between real and simulated observations (O-B) using chemical information from reference profiles (black line) and from *in situ* profiles (red line) with respect to IASI (a) and CrIS (b) monitored channels for radiosonde from 2018-04-17 at 10 UTC.

*in situ* chemical profiles (SONDE) are small over all IASI (a) and CrIS (b) spectrum. However, there are significant differences for ozone-sensitive channels with (O-B) values closer to zero when using the reference ozone. This could be explained by

5    a smaller accuracy of ozone measurement with ozone-sondes than with the AMULSE instrument. Another possibility is the existence of other biases (eg. depending on the scan position) which are compensated for by bias of the opposite sign in the case of REF. The same experiments were performed on other radiosonde data with generally the same results (not shown here). Thus, *in situ* chemical profiles allow us to validate the quality of our simulations essentially for channels sensitive to $CO_2$ and $CH_4$. However, it is more difficult to accurately simulate ozone-sensitive channels. Indeed, ozone differs from $CO_2$ and $CH_4$

10    because it has a very high spatial and temporal variability.



## 5.2 Comparison between *a priori*, *retrieval* and *in situ* chemical profiles

*In situ* chemical profiles can directly be used as verification data for modeled profiles or retrievals from satellite sensors. In this section, two possible usages are illustrated on APOGEE data.

### 5.2.1 Forecasts of atmospheric composition

One of the possible use of these *in situ* profiles of $CO_2$, $CH_4$ and $O_3$ is to use them as comparison data to the *a priori* chemical profiles from the Chemistry Transport Models (CTM). As part of our study, we compared our measured data against the profiles extracted from the CTM MOCAGE and C-IFS. These two models differ in that the global version fo the MOCAGE model does not assimilate any observation in operation, while C-IFS assimilates Level-2 products from several instruments such as MLS, OMI, SBUV-2, GOME-2A/2B, IASI, MOPITT, OMPS and PMAp (Basart et al., 2018). C-IFS provides analyses and 5-day forecasts of atmospheric composition at regional and global scales in near-real time. MOCAGE is an off-line global three-dimensional chemistry transport model (Guth et al., 2016). It provides the time evolution of the chemical air composition from the surface to the stratosphere. MOCAGE is used for operational daily forecasts and also for research studies. Thus, we extracted the profiles *a priori* from $CH_4$ and $O_3$ from the CTM MOCAGE and C-IFS for our case study of the radiosonde from 2018-04-17 to 10 UTC. These profiles could therefore be compared to the *in situ* and reference profiles available in RTTOV. We have shown in Figure 11 the ozone (a) and methane (b) *in situ* profiles in black line compared to $O_3$ and $CH_4$ *a priori* profiles from CAMS in red line and MOCAGE in blue line and compared to $O_3$ and $CH_4$ reference profiles in dotted.

In Figure 11.a we notice that the reference ozone profile is relatively in good agreement with the *in situ* profile except in the upper troposphere between 300 and 150 hPa. The ozone values of the reference profile are slightly higher than the *in situ* profile in the lower troposphere between 1000 and 350 hPa and in the lower stratosphere between 60 and 20 hPa. Then, we observe that the *a priori* profile of ozone from MOCAGE is very close to the profile measured over the entire lower stratosphere. The MOCAGE ozone profile follows the profile measured in the upper troposphere between 500 and 80 hPa, with lower values. The MOCAGE ozone values are overestimated compared to the profile measured in the lower troposphere between 1000 and 500 hPa. Finally, the ozone profile from CAMS is in very good agreement with the profile measured in the lower troposphere between 1000 and 450 hPa and in the lower stratosphere between 45 and 15 hPa. However, CAMS ozone profile overestimates the values compared to the profile measured over a large part of the atmosphere between 500 and 50 hPa. Overall, the different ozone profiles are more or less close to the measured profile. The CTMs are able to simulate the shape of ozone profiles relatively well with good values for MOCAGE in the stratosphere and in the lower troposphere for CAMS. The reference profile seems to be a good compromise on this particular case.

Methane simulation is a much more difficult task in the field of CTMs. Indeed, there are major scientific questions about the increase in atmospheric methane concentration and its hypothetical sources. That is why assimilation into CTMs for this species is very useful. We notice again in Figure 11.b that the reference methane profile is rather good with the profile measured





between 1000 and 70 hPa. Indeed, above 70 hPa, there is a faster decrease in the measured concentration than the values of the reference profile. This decrease is simulated by the profile from MOCAGE but with a large underestimation over the entire profile of about 0.5 ppmv. This difference can be explained by a missing source in the model and/or too many OH molecules that are the main methane sink. Assimilation can solve this problem, so there is a very good agreement between CAMS methane

profile and the measured profile in the lower troposphere between 1000 and 300 hPa. Then the values of the CAMS profile are underestimated compared to the measured profile between 300 and 45 hPa.

Such *in situ* chemical profiles can be very useful in assessing the quality of simulations of atmospheric composition from CTMs, especially if they can be made on a regular basis within a network. For our study case, we note the rate and concentration

of ozone are relatively well simulated by MOCAGE and CAMS, while methane is largely underestimated in MOCAGE and very well represented in the troposphere by CAMS.

### 5.2.2   Ozone retrieval from 1D assimilation

Profiles of various atmospheric compounds can beretrieved from satellite data. Among them, hyperspectral infrared sounders like IASI have the ability to be used both in NWP models and for atmospheric composition purposes. Which could be used in

combining meteorological and chemistry transport models to achieve a so-called coupled assimilation. A precursor way is to add some chemical variable to the control variable of NWP models. This is the case, for example, for ozone in the IFS model of ECMWF, which assimilates 16 IASI ozone-sensitive channels in operational near-real time providing ozone analysis (Han and McNally, 2010). Such an approach can also be used at the pixel to retrieve profiles. Indeed, (Coopmann et al., 2018) study shows that the assimilation of 15 ozone sensitive IASI channels makes it possible to simultaneously improve temperature,

humidity and ozone analyses. In both cases, it is also important to assess the quality of ozone analyses using independent comparative data. For ozone, this can be done almost in any region around the world since there are several stations that perform radiosondes measuring ozone. However, if methane or carbon dioxide analyses are to be retrieved, *in situ* vertical profiles of these compounds are much sparse. This is why the data measured by the AMULSE instrument can be valuable when these compounds are reproduced by meteorological models. As part of the APOGEE measurement campaign, we carried out several

assimilation experiments in a simplified one-dimensional framework (1D-Var) where we retrieved ozone. Thus, the measured of *in situ* ozone profiles could be used as comparative data to evaluate the quality of our restitutions. In this study we conducted these experiments using observations from IASI and CrIS, focusing on simultaneous retrieval of temperature, humidity and ozone, taking ozone background/*a priori* from MOCAGE and CAMS. Soundings of 2017-06-01 to 10 UTC, 2017-07-04 to 02 UTC, 2017-07-04 to 09 UTC and 2017-07-04 to 12 UTC are selected for that case study. The methodology and techniques

used in this study are the same as those used in (Coopmann et al., 2018).

To retrieve ozone using observations from IASI, we assimilated 123 channels used in operations at Météo-France plus 15 ozone-sensitive channels, using a full observation error covariance matrix diagnosed from the method described by (Desroziers et al., 2005). Based on CrIS observations, we assimilated 68 channels used in operations at Météo-France and 14 ozone-





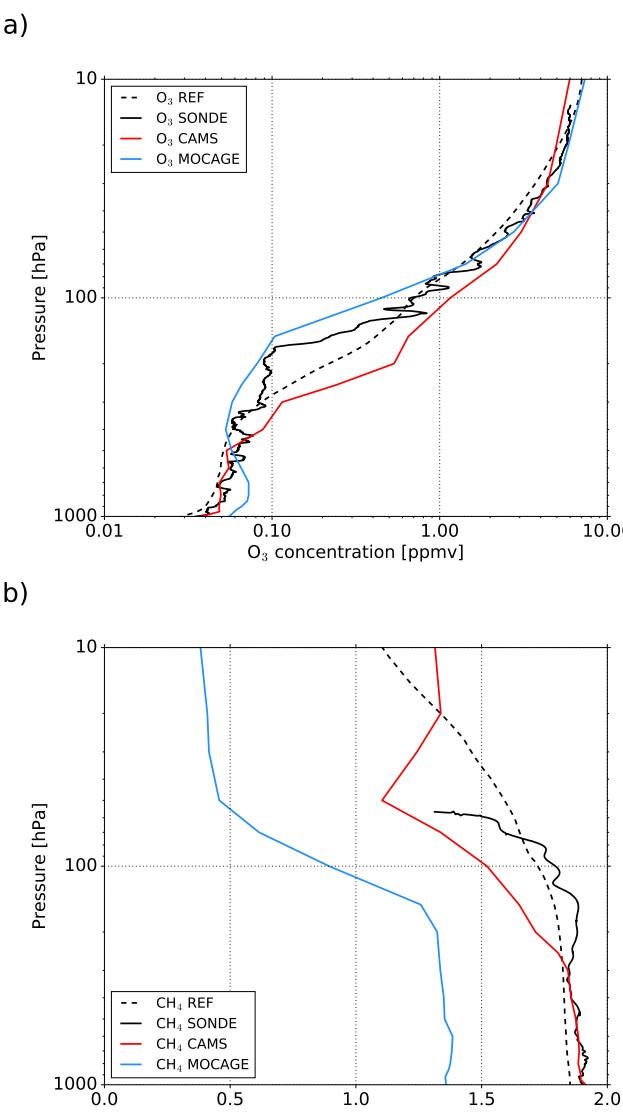

**Figure 11.** Ozone (a) and methane (b) *in situ* profiles in black line compared to $O_3$ and $CH_4$ *a priori* profiles from CAMS in red line and MOCAGE in blue line and compared to $O_3$ and $CH_4$ reference profiles in dotted for radiosonde from 2018-04-17 at 10 UTC.

sensitive channels, using a diagonal observation error covariance matrix whose values are derived from the operational settings at Météo-France. IASI and CrIS observations were collocated with radiosondes. The ozone background errors available in the background error covariance matrix were calculated using ozone-sondes based on the method described in the work of (Coopmann et al., 2018). Finally, the thermodynamic *a priori* profiles and surface parameters were extracted from the global

5 model ARPEGE and were extracted at the same coordinates and time as the radiosondes, same for the ozone *a priori* profiles





from MOCAGE and CAMS.

Figure 12 represent the ozone *in situ* profile in black compared to $O_3$ *a priori* profile from MOCAGE in red and $O_3$ *retrieval* profile from CrIS in blue for radiosonde from 2017-07-04 at 02 UTC (a) and 2017-07-04 at 12 UTC (c) and ozone *in situ* pro-

file in black compared to $O_3$ *a priori* profile from CAMS in orange and $O_3$ *retrieval* profile from CrIS in cyan for radiosonde from 2017-07-04 at 02 UTC (b) and 2017-07-04 at 12 UTC (d). In Figure 12.a, we notice that overall the MOCAGE *a priori* profile underestimates the ozone concentration compared to the measured profile except in the lower troposphere and above 25 hPa. In this case, the ozone *retrieval* profile is very close to the measured profile. In Figure 12.b, the CAMS *a priori* profile is generally closer to the measured profile than the MOCAGE *a priori* profile. However, there are some overestimation of CAMS

ozone concentration between 300 - 200 hPa and 150 - 50 hPa. We note that with this *a priori* profile, the ozone *retrieval* profile is less modified than previously. In the same way as the previous radiosonde, we notice in Figure 12.c that the MOCAGE *a priori* profile underestimates the ozone concentration compared to the measured profile except in the lower troposphere. The ozone *retrieval* profile is again of very good quality. Finally, we note in Figure 12.d that CAMS's *a priori* profile underestimates the concentration of ozone in the lower troposphere compared to the measured profile and overestimates it between

500 - 20 hPa. In this experiment, the ozone *retrieval* profile is very close to the measured profile except in the lower troposphere.

Figure 13 represent ozone *in situ* profile in black compared to $O_3$ *a priori* profile from MOCAGE in red and $O_3$ *retrieval* profile from IASI in blue for radiosonde from 2017-06-01 at 10 UTC (a) and 2017-07-04 at 09 UTC (c) and ozone *in situ* profile in black compared to $O_3$ *a priori* profile from CAMS in orange and $O_3$ *retrieval* profile from IASI in cyan for radiosonde from

2017-06-01 at 10 UTC (b) and 2017-07-04 at 09 UTC (d). Figure 13.a shows that the *a priori* from MOCAGE underestimates the ozone concentration compared to the measured profile except in the lower troposphere. This experiment makes it possible to retrieve a profile very close to the measured profile except above 50 hPa where the latter remains close to the *a priori*. In Figure 13.b, the *a priori* from CAMS is generally close to the measured profile except at the UTLS where there is an overestimation of the ozone concentration. It can be seen that with this *a priori* profile, the *retrieval* profile approaches the measured profile and

shows the same structure as the measured profile in the UTLS. Then Figure 13.c shows that the *a priori* profile from MOCAGE underestimates the ozone concentration compared to the profile measured in this case except in the lower troposphere and above 25 hPa. The *retrieval* profile is also of good quality. Finally, we note in Figure 13.d that the *a priori* profile from CAMS underestimates ozone concentration in the lower troposphere compared to the measured profile and overestimates it by 150 - 40 hPa. This last experiment allows to retrieve an ozone profile very close to the measured profile except between 300 - 150

hPa.

## 6 Conclusions

This article shows the interest and possibilities of the AMULSE instrument for the fast and accurate measurement of $CO_2$, $CH_4$ and $H_2O$ in the atmosphere. The APOGEE measurement campaign was an opportunity to highlight the various benefits



**Figure 12.** Ozone *in situ* profile in black compared to $O_3$ *a priori* profile from MOCAGE in red and $O_3$ *retrieval* profile from CrIS in blue for radiosonde from 2017-07-04 at 02 UTC (a) and 2017-07-04 at 12 UTC (c). Ozone *in situ* profile in black compared to $O_3$ *a priori* profile from CAMS in orange and $O_3$ *retrieval* profile from CrIS in cyan for radiosonde from 2017-07-04 at 02 UTC (b) and 2017-07-04 at 12 UTC (d).

of the data from the AMULSE instrument for different research projects: satellite validation, model comparison, verification data, etc. The measurements will continue with monthly radiosoundings on the Reims site to get more consistent data sets.

The prime aim of this study was to assess the sensitivity of infrared satellite observations to chemical information. The objec-
5    tive was to use carbon dioxide, methane and ozone profiles measured as part of the APOGEE measurement campaign to validate the quality of IASI and CrIS observation simulations. The *in situ* profiles were measured for ozone using an electrochemical cell and the $CO_2$ and $CH_4$ profiles using the AMULSE instrument. These instruments were placed under meteorological balloons to produce vertical profiles up to 30 km in altitude. Experiments for one case study showed us that infrared observations



**Figure 13.** Ozone *in situ* profile in black compared to $O_3$ *a priori* profile from MOCAGE in red and $O_3$ *retrieval* profile from IASI in blue for radiosonde from 2017-06-01 at 10 UTC (a) and 2017-07-04 at 09 UTC (c). Ozone *in situ* profile in black compared to $O_3$ *a priori* profile from CAMS in orange and $O_3$ *retrieval* profile from IASI in cyan for radiosonde from 2017-06-01 at 10 UTC (b) and 2017-07-04 at 09 UTC (d).

are extremely sensitive to changes in chemical concentrations. The RTM RTTOV model allowed us to perform several simulations of infrared observations under several configurations of chemical *a priori* profiles provided as input. This is how we were able to highlight differences in simulation between the use of the *in situ* and reference profiles. However, despite these differences, we have shown little impact in differences between the real and simulate observations using either the reference

5   chemical profiles or the *in situ* profiles except for ozone. Indeed, the latter reacts differently in the atmosphere and is highly variable compared to $CO_2$ and $CH_4$. The data measured by the AMULSE instrument are valuable to validate the quality of our simulations, which is essential for NWP models.

Secondly, the measured profiles in the APOGEE framework were also valuable for comparison with the chemical profiles derived from the CTMs. Thus, we extracted ozone and methane profiles from the MOCAGE and CAMS CTMs for a case study. This comparison with the measured profiles showed us that the CTMs represent atmospheric ozone rather well, especially in the stratosphere for MOCAGE and in the troposphere for CAMS. Having good chemical concentrations in the troposphere

for CAMS is important as it also provides air quality forecasts. We also showed that methane is very well simulated by the CTM CAMS in the troposphere but underestimated by 0.5 ppmv by the CTM MOCAGE. The difference is mainly due to the fact that CAMS's C-IFS model assimilates level 2 products compared to MOCAGE which does not assimilate anything. Once again, we can see the importance of these *in situ* profiles for the comparison and validation of chemical predictions from CTMs.

Finally, a method to obtain more accurate chemical profiles was used. In this case, ozone profiles were retrieved using one-dimensional assimilation (1D-Var) using observations from IASI and CrIS and *a priori* ozone profiles from MOCAGE and CAMS for 4 radiosondes. The *in situ* profiles have allowed us to highlight very encouraging results since, the method used in this study allows us to retrieve ozone profiles very close to the measured ozone profiles, particularly in the UTLS which remains an area difficult to model by CTMs. We have also shown that some retrieved ozone profiles simulate very well the

complex structures that some measured ozone profiles generated by isentropic transport in the UTLS can have.

AMULSE regularly evolves in terms of performance, weight and number of detectable gas molecules. Different perspectives are possible, such as the measurement of other molecular species of atmospheric interest in captive balloons or weather balloons for measurements in the stratosphere.

*Acknowledgements.* This research based on data from the APOGEE measurement campaign has been conducted within the framework of O. Coopmann's PhD thesis, which funded by CNES (Centre National d'Études Spatiales) and the Région Occitanie. Thanks to Michel Ramonet (from LSCE) and his team for allowing us to perform the measurements on their ICOS site.

*Code availability.* Codes of the Radiative Transfer Model RTTOV and the uni-dimensional data assimilation system 1D-Var used is this study are all available on (https://www.nwpsaf.eu/site/software/rttov/download/).

*Data availability.* IASI data are available from EUMETSAT or AERIS: (https://www.aeris-data.fr/). Copernicus Atmosphere Monitoring Service data are available from: (https://atmosphere.copernicus.eu/catalogue#/) implemented by ECMWF as part of The Copernicus Programme. Model data are available upon request.

*Competing interests.* The authors declare that they have no conflict of interest.





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
