# Peer review of "The evolution of AMULSE (Atmospheric Measurements by Ultra-Light Spectrometer) and its interest in atmospheric applications. Results of the Atmospheric Profiles Of GreenhousE gasEs (APOGEE) weather balloon release campaign for satellite retrieval validation"

_Atmospheric Measurement Techniques, 2019_

## Referee Comment (RC1) · Anonymous Referee #1 · 15 Nov 2019

OVERVIEW: This paper describes the ongoing development of a novel new greenhouse gas (GHG) measurement system designed for deployment on standard meteorological balloons using an ultralight (< 3 kg) mid-IR spectrometer. Because in situ trace gas profile measurements are of high value for atmospheric chemistry, transport models, climate change studies, and satellite validation, this topic is of interest to, and

suitable for, publication in AMT.

RECOMMENDATIONS:

Reconsidered after major revisions.

GENERAL COMMENTS

1) While the paper's topic is of high interest for reasons cited above, major revisions will be needed to bring the manuscript to publication quality.

2) Certain sections (detailed more below) would benefit from additional expository, where other sections are questionable as to the scope of a single paper. Additional content will be necessary to link the work with the large quantity of current and previous related work.

3) Linkage to current/previous work (e.g., other trace gas measurement systems and satellite validation) is essential as the paper seeks to identify itself as of something of "interest in atmospheric applications," but then falls woefully short of pointing out specific applications with appropriate citations to existing/previous work (some given below). Previous work includes retrieval algorithm validation work from the AIRS and CrIS instruments and other in situ greenhouse gas measurements. The AIRS instrument isn't even mentioned in the submitted paper, nor campaigns such as HIPPO or ATom, nor well-established networks such as TCCON, which is a glaring oversight given the paper's original stated goal of "interest in atmospheric applications and satellite validation." This needs to be corrected in the revision.

4) There are numerous issues with grammar throughout (e.g., spelling errors, problems with singular/plural usage, etc.); I have identified out some corrections below, but not all of them.

SPECIFIC COMMENTS

1) Title: IMO the title could be shortened (the entire second part could be deleted) and

modified. A suggestion is as follows: "The development of the Atmospheric Measurements by Ultra-Light Spectrometer (AMULSE) greenhouse gas profiling system and its interest in atmospheric applications"

2) P. 1

a) Line 5: Rewrite "under weather and tethered balloons" as "via standard weather and tethered balloons"

b) Line 8: replace "tethered balloon campaign and for a balloon campaign" with "two experiments" 3) P. 2

a) Line 7-8: rewrite as "During the last decades, evidence has been accumulated that this climate change is directly related to the human activates" and include the 2019 IPCC Report and 2019 AMS State of the Climate citations.

b) Line 21: Here and elsewhere, plural/singular usage needs to be corrected. I do not have time point out every occurrence here in a formal review – it is systematic throughout the document and needs to be corrected. In this case, "many informations" should be replaced simply with "information" – "information" is already both singular and plural – there is no such word as "informations". 4) P. 4

a) Line 26: "meters' " should be simply "meters".

b) Line 27: Pertaining to plural usage, replace "lots of preparations" with "a lot of preparation"

5) P. 5

a) Line 2: "The specificity of the balloons, to be able to access the profiles" – meaning not clear.

b) Line 11: "resolution of few meters" – needs to be more quantitative, e.g., "1-5 meters" (or something like that)

c) Line 21: "atmospheric sensing" is much too vague – rewrite as "measuring atmospheric gas concentrations"

d) Line 28: "Lambert Beer's law" is usually referred to either as "Beer-Lambert law" or "Beer's Law"

6) P. 6

a) Line 13: What is meant by "meteorological fields"? Such terminology usually refers to an analysis or model of a particular set of state parameters, but I don't think that's what you're talking about with an Iridium module.

b) Line 14: "computer fixed at 400 m" – what does that mean or how is it relevant?

c) Line 16: Capitalize acronym "AMULSE"

d) Line 18, Table Caption: Delete "This table illustrates" and begin simply with "Evolution (between 2015 and 2018. . .)

7) P. 7

a) Lines 1-2: Please give the fractional differences (%) as well.

b) Figure 1 Caption: capitalize AMULSE and don't refer to the insets as (a), (b), (c), which usually simply refer to the panels of a multi-panel figure. Instead simply refer to them as "insets".

8) P. 8

a) Line 5: Here and elsewhere, replace "captive balloon" with "tethered balloon" – "captive balloon" is not a standard terminology.

b) Line 8: Here and elsewhere, I suggest replacing the word "campaign" with "experiment", based upon the descriptions of said experiments. A "campaign" usually refers to a dedicated mission that deploys single or multiple moving platform aircraft (e.g., ATom or CalWater), ships, or a dedicated observing network spread over an area and

working in coordination with one another over an intensive observing period. Perhaps the Authors haven't fully described their mission or I have misunderstood, but what they describe otherwise sounds more a like an experiment.

c) Line 9: replace "spatial" with "vertical" – spatial resolution refers to horizontal resolution.

d) Line 10: to what point spacing are the data interpolated?

e) Lines 10-13: Need to comment/discuss the boundary layer evolution shown on the figure, or otherwise delete the figure.

f) Figure 2: The H2O is given in %, but % what? I presume it's not RH.

9) P. 9

a) Line 1: Reiterate what APOGEE stands for, and more details on where and when it was conducted.

b) Line 9: What does "GSMA" stand for?

c) Line 10: sentence needs period.

10) P. 10

a) Figure 3 caption: Rewrite "connected in order to send" with "connected which measures and transmits"

b) Figure 4: Is this the "GSMA" site? More details are needed in the caption.

11) P. 11

a) Table 2: Either insert commas "," in the altitude numbers (e.g., 19,121) or rewrite in km (e.g., 19.121).

b) Line 3: "photosynthesis phenomenon"? How does photosynthesis "enrich" CO2?

c) Line 6: insert "the year" before 2020 for clarity.

12) P. 12

a) Section 5.1.2 Radiative Transfer Model: To my knowledge, greenhouse gases (or greenhouse gas channels) are minimally assimilated into NWP models – these models assimilate thermodynamic sounder channels (e.g., temperature/moisture) for forecasting. So it is not clear to me what the ultimate relevance of this section is to the paper, which ought to be more focused on the new (and novel) in situ gas sampling system.

b) Line 6: "a prior" should be "a priori"

c) Line 11: "The differences between observations and simulations are called innovations." – I am unfamiliar with the terminology "innovations".

d) Line 15: Capitalize TIROS.

13) P. 13

a) Line 14: insert "near the surface" between "high sensitivity" and "to window channels"

b) Line 15: Here and elsewhere: Use decimal points to separate the unit and tenth decimal places – at first I was completely confused when I saw 1.080 to 1.150 cm-1, thinking these were near-IR to microwave channels when talking about an IR instrument. Please rewrite simply as "1080 to 1150 cm-1")

14) P. 14

a) Last line: What does "ARPEGE" stand for?

15) P. 15

a) Last line: It should also be noted that there is a temporal difference in the overpass times between IASI and CrIS. Also, the Author should always be explicit as to what satellites they're talking about. For IASI is it Metop-A, -B or –C? For CrIS is it Suomi NPP or NOAA-20?

16) P. 16

a) Equation 1: These equations could be expressed better. Why "EXP"? Is it possible to simplify the subscripts by abbreviating them to a simply letter (e.g., replace "simul" with "s", "REF" with "r" or "b" for background, "AMULSE" with "o" for "obs", etc.) Also, why are three variables in boldface?

17) P. 17

a) Figure 9 caption: What does ",rp" mean?

b) Lines 5-6: Please explain the relevance to the current paper.

18) P. 18

a) Lines 8-10: It is not clear what relevance this has. Are simulations usually "validated" in this manner? Usually what is done is that radiative transfer model (RTM) "obs minus calc" are subjected to empirical bias correction "tuning" using observations such as this (where the RTM is then later used in retrieval algorithms), but that's not "validation."

19) P. 19: nothing

20) P. 20:

a) Line 14: include "AIRS, CrIS" along with IASI; Combine sentence fragments by replacing "." with "," before "which".

b) Line 17: insert "level" after "pixel" and insert "or from cloud-cleared spectra as done by the AIRS and NUCAPS systems (Susskind et al., 2003; Smith and Barnet, 2019)." (References provided below)

c) Line 20: Replace "ozone analyses" with "ozone retrievals"

d) Line 23: Insert citation "(Nalli et al. 2018)" for the ozone retrieval validation connection.

e) Line 26: What does "restitutions" mean?

21) P. 21

a) Line 2: What radiosondes?

22) P. 22: The Authors should provide a bit more information on their 1-D VAR retrieval algorithm and then should relate their results using this algorithm with results from established operational algorithms such as NUCAPS (as in the Nalli et al. reference) and/or AIRS.

23) P. 23

a) Figure 12: Because the Authors aren't even showing AMULSE measurements in these figures, it's important that they related it to their AMULSE work and also tie it back to other work on the subject of ozone validation using ozonesondes.

b) Figure 12d: Fix typo "Radiosondge" should be "Radiosonde"

24) P. 24

a) Figure 13 Caption: Simplify by rewriting as "As Figure 12, except for IASI. . ."

b) How is "validating the quality of our simulations essential for NWP models"?

25) P. 26

a) Line 13: Please provide full citation.

REFERENCES (not all-inclusive – please feel free to include more if you have them, including specifically references to other in situ measurements such as HIPPO, ATom)

Blunden, J. and D. S. Arndt, Eds., 2019: State of the Climate in 2018. Bull. Amer. Meteor. Soc., 100 (9), Si–S305, doi:10.1175/2019BAMSStateoftheClimate.1.

Chahine et al., 2006: AIRS: Improving weather forecasting and providing new data on greenhouse gases, Bull. Amer. Meteorol. Soc, 87, 911-926.

IPCC, 2014: Climate Change 2014: Synthesis Report. Contribution of Working Groups

I, II and III to the Fifth Assessment Report of the Intergovernmental Panel on Climate Change [Core Writing Team, R.K. Pachauri and L.A. Meyer (eds.)]. IPCC, Geneva, Switzerland, 151 pp.

Nalli, N. R., et al., 2018: Validation of atmospheric profile retrievals from the SNPP NOAA-Unique Combined Atmospheric Processing System. Part 2: Ozone, IEEE Trans. Geosci. Remote Sens., 56(1), 598-607, doi:10.1109/TGRS.2017.2762600.

Smith, N. and C.D. Barnet, 2019: Uncertainty characterization and propagation in the community long-term infrared Microwave Product System (CLIMCAPS), Remote Sens., 11, 1227, doi:10.3390/rs11101227.

Susskind, J., C.D. Barnet, and J.M. Blaisdell, 2003: Retrieval of atmospheric and surface parameters from AIRS/AMSU/HSB data in the presence of clouds, IEEE. Trans. Geosci. Remote Sens., 41, 390-409.

---

## Referee Comment (RC2) · Anonymous Referee #2 · 3 Jan 2020

General Comments 1) This paper should be reconsidered after major revisions. 2) There are sections of the paper that focus solely on IASI analysis and then shoehorn in the fact that the analysis was also done with CrIS. It would be beneficial to add more details on the analysis done with CrIS. A couple of the sections that need more information are mentioned in the specific comments. 3) There are many grammatical

mistakes throughout that need fixing. I compiled a list of the ones I found but there are certainly more. 4) Overall, the structure of the paper is good.

Specific comments 1) Many acronyms are used without being spelled out in the abstract (CNRM, AMULSE, APOGEE, IASI, CrIS, NWP, RRTOV, MOCAGE, CAMS) 2) P8. Figure 2: What does the % of H2O represent? 3) P11. Line 3: The photosynthesis phenomenon enriching CO2 should have a reference 3) In section 5.1.3 and 5.2.2, the authors should add more information on CrIS. CrIS seems to be mentioned off hand while all of the analysis is done with IASI. 4) In equation 1 in section 5.1.5 exp should be rewritten as either exp(O3) or O3=ln(.....). The superscript simul could also be shortened to just sim.

Technical corrections P1. Line 5-6: I recommend splitting this into two sentences. Maybe "The plug and play instrument is compact, robust, cost-effective, and autonomous. The instrument also has a low power consumption and is non-intrusive." P2. Line 7-8: rewrite to "The climate of the earth is currently changing quickly. Recently, evidence has accumulated showing that this climate change is directly related to human activities." Line 9: change "involve" to "create" Line 10: change "of" to "on" Line 10: remove "on Earth" Line 20: remove "the" before CH4 Line 22: change "many informations" to "information" Line 25: Define the RRTOV acronym P4. Line 26: change "meters'" to "meters" Line 27: change "lots of preparations" to "a lot of preparation" P5. Line 2: change "This specificity of the balloons, to be able to access the pofiles," to "The balloons' ability to access the profiles" P6. Line 9: close the parentheses Line 10: change "The spectrometer weights is lower than" to "The spectrometer weighs less than" Line 13: change "send various datas" to "sends various data" Line 14: computer fixed 400m in relation to what? Line 15: capitalize AMULSE Line 23: change "at" to "a" change "about" to "for" Line 25: capitalize AMULSE P7. Figure 1: capitalize AMULSE, remove "of (a) CO2,(b) CH4 and (c) H2O" P9. Line 3: replace "the" with "our" Line 9: change "balloon" to "balloons" Line 10: add period at the need of this sentence Line 18: change "Balloon" to "The balloon" Line 20: change "Balloon" to "The balloon" Line 20:

change "cutted" to "cut" Line 29: make hypothesis plural P11. Move the table 2 caption down a line Line 6-7: change "prospects for 2020 is to make water vapour measurements by laser diode spectrometry to have a better accuracy" to "goals for 2020 is to make water vapour measurements by laser diode spectrometry more accurate" P12. Line 6: change "a prior" to "a priori" P13. Line 15,19,20, and 21: Either use commas or no separation between the thousand and hundred place in numbers instead of decimal points. (i.e. 1,000 or 1000) P16. Need to fix decimal points on this page as well. Line 4 and 5: what does "rp" mean Why did CO2 and CH4 become italicized midway through this page P17. Figure 9: If ",rp" stands for reference profiles you need a better way of conveying this Line 5: add "to" between "CO2" and "have"

P18. Line 8: change "validate to "verify" and remove "essentially" P19. Line 5: Remove "of the" P20. Line 13: add space to "beretrieved" Line 14: change ". Which" to ", which" Line 25: Remove "of" add the start of the sentence. I do not think "restitutions" is the right word here. P22. Last 2 lines: Remove italics on chemical names P23. Figure 12 panel d: change "radiosondage" to "radiosonde"

---

## Author Comment (AC1) · 5 Feb 2020

Review#1

The evolution of AMULSE (Atmospheric Measurements by Ultra-Light Spectrometer) and its interest in atmospheric applications. Results of the Atmospheric Profiles Of

GreenhousE gasEs (APOGEE) weather balloon release campaign for satellite retrieval validation

Anonymous Referee #1

OVERVIEW: This paper describes the ongoing development of a novel new greenhouse gas (GHG) measurement system designed for deployment on standard meteorological balloons using an ultralight (< 3 kg) mid-IR spectrometer. Because in situ trace gas profile measurements are of high value for atmospheric chemistry, transport models, climate change studies, and satellite validation, this topic is of interest to, and suitable for, publication in AMT.

Thank you very much for your interest in our work. I warmly thank the Reviewer for his valuable comments which enabled us to improve this article.

RECOMMENDATIONS:

Reconsidered after major revisions.

GENERAL COMMENTS

1) While the paper's topic is of high interest for reasons cited above, major revisions will be needed to bring the manuscript to publication quality.

2) Certain sections (detailed more below) would benefit from additional expository, where other sections are questionable as to the scope of a single paper. Additional content will be necessary to link the work with the large quantity of current and previous related work.

3) Linkage to current/previous work (e.g., other trace gas measurement systems and satellite validation) is essential as the paper seeks to identify itself as of something of "interest in atmospheric applications," but then falls woefully short of pointing out specific applications with appropriate citations to existing/previous work (some given

below). Previous work includes retrieval algorithm validation work from the AIRS and CrIS instruments and other in situ greenhouse gas measurements. The AIRS instrument isn't even mentioned in the submitted paper, nor campaigns such as HIPPO or ATom, nor well-established networks such as TCCON, which is a glaring oversight given the paper's original stated goal of "interest in atmospheric applications and satellite validation." This needs to be corrected in the revision.

I agree, a link with other chemical measurement instruments (e.g. AIRCORE) and with other measurement campaigns and satellite validation is missing. Changes have been made in the article to correct this deficiency.

4) There are numerous issues with grammar throughout (e.g., spelling errors, problems with singular/plural usage, etc.); I have identified out some corrections below, but not all of them.

SPECIFIC COMMENTS

1) Title: IMO the title could be shortened (the entire second part could be deleted) and modified. A suggestion is as follows: "The development of the Atmospheric Measurements by Ultra-Light Spectrometer (AMULSE) greenhouse gas profiling system and its interest in atmospheric applications"

Corrected

2) P. 1 a) Line 5: Rewrite "under weather and tethered balloons" as "via standard weather and tethered balloons"

Corrected

b) Line 8: replace "tethered balloon campaign and for a balloon campaign" with "two experiments"

Corrected

3) P. 2 a) Line 7-8: rewrite as "During the last decades, evidence has been accumulated

that this climate change is directly related to the human activates" and include the 2019 IPCC Report and 2019 AMS State of the Climate citations.

b) Line 21: Here and elsewhere, plural/singular usage needs to be corrected. I do not have time point out every occurrence here in a formal review – it is systematic throughout the document and needs to be corrected. In this case, "many informations" should be replaced simply with "information" – "information" is already both singular and plural – there is no such word as "informations".

Corrected

4) P. 4 a) Line 26: "meters' " should be simply "meters".

Corrected

b) Line 27: Pertaining to plural usage, replace "lots of preparations" with "a lot of preparation"

Corrected

5) P. 5 a) Line 2: "The specificity of the balloons, to be able to access the profiles" – meaning not clear.

b) Line 11: "resolution of few meters" – needs to be more quantitative, e.g., "1-5 meters" (or something like that)

Corrected

c) Line 21: "atmospheric sensing" is much too vague – rewrite as "measuring atmospheric gas concentrations"

Corrected

d) Line 28: "Lambert Beer's law" is usually referred to either as "Beer-Lambert law" or "Beer's Law"

Corrected

6) P. 6 a) Line 13: What is meant by "meteorological fields"? Such terminology usually refers to an analysis or model of a particular set of state parameters, but I don't think that's what you're talking about with an Iridium module.

Corrected

b) Line 14: "computer fixed at 400 m" – what does that mean or how is it relevant?

Corrected

c) Line 16: Capitalize acronym "AMULSE"

Corrected

d) Line 18, Table Caption: Delete "This table illustrates" and begin simply with "Evolution (between 2015 and 2018. . .) Corrected

7) P. 7 a) Lines 1-2: Please give the fractional differences (%) as well.

Corrected

b) Figure 1 Caption: capitalize AMULSE and don't refer to the insets as (a), (b), (c), which usually simply refer to the panels of a multi-panel figure. Instead simply refer to them as "insets".

Corrected

8) P. 8 a) Line 5: Here and elsewhere, replace "captive balloon" with "tethered balloon" – "captive balloon" is not a standard terminology.

Corrected

b) Line 8: Here and elsewhere, I suggest replacing the word "campaign" with "experiment", based upon the descriptions of said experiments. A "campaign" usually refers to a dedicated mission that deploys single or multiple moving platform aircraft (e.g., ATom or CalWater), ships, or a dedicated observing network spread over an area and working in coordination with one another over an intensive observing period. Perhaps

the Authors haven't fully described their mission or I have misunderstood, but what they describe otherwise sounds more a like an experiment.

Corrected

c) Line 9: replace "spatial" with "vertical" – spatial resolution refers to horizontal resolution.

Corrected

d) Line 10: to what point spacing are the data interpolated? X, Y, Z : [20, 100, 100]

Corrected

e) Lines 10-13: Need to comment/discuss the boundary layer evolution shown on the figure, or otherwise delete the figure.

Corrected

f) Figure 2: The $H_2O$ is given in %, but % what? I presume it's not RH. Corrected

9) P. 9 a) Line 1: Reiterate what APOGEE stands for, and more details on where and when it was conducted.

APOGEE (Atmospheric Profiles Of GreenhousE gasEs) is the name of a French scientific project which has funded this project. Date included

b) Line 9: What does "GSMA" stand for? Corrected

c) Line 10: sentence needs period.

Corrected

10) P. 10 a) Figure 3 caption: Rewrite "connected in order to send" with "connected which measures and transmits"

Corrected

b) Figure 4: Is this the "GSMA" site? More details are needed in the caption. Corrected

11) P. 11 a) Table 2: Either insert commas "," in the altitude numbers (e.g., 19,121) or rewrite in km (e.g., 19.121).

Corrected

b) Line 3: "photosynthesis phenomenon"? How does photosynthesis "enrich" CO2?

During the day, plants take advantage of photosynthesis. They release more oxygen than CO2. During the night, there is no more photosynthesis, the plants breathe. They only emit CO2.. Corrected Âń plant respiration Âż

c) Line 6: insert "the year" before 2020 for clarity.

Corrected

12) P. 12 a) Section 5.1.2 Radiative Transfer Model: To my knowledge, green-house gases (or greenhouse gas channels) are minimally assimilated into NWP models – these models assimilate thermodynamic sounder channels (e.g., tempera-ture/moisture) for forecasting. So it is not clear to me what the ultimate relevance of this section is to the paper, which ought to be more focused on the new (and novel) in situ gas sampling system.

Currently, most of the channels used in data assimilation for Numerical Weather Pre-diction are the CO2 sensitive channels in the infrared spectrum to retrieve temperature profiles, H2O sensitive to retrieve humidity profiles and some window channels. This is why a realistic consideration of a CO2 prior profile is preferable at the input of Radia-tive Transfer Models in order to improve the quality of the simulated channels sensitive to this greenhouse gas as shown in the [Engelen et al., 2014] work. Finally, several meteorological centres are beginning to assimilate channels sensitive to ozone and methane for example (this is the case of the IFS model at the ECMWF) in order to extract information on temperature and humidity, but also on the molecules themselves in the case where the latter is also modified during the assimilation process.

b) Line 6: "a prior" should be "a priori"

corrected

c) Line 11: "The differences between observations and simulations are called innovations." – I am unfamiliar with the terminology "innovations".

Innovation is the term used in the data assimilation community [Ide et al., 1999] can also be called first-guess departure. Âń innovation Âż has been remplaced by Âń first-guess departure Âż.

d) Line 15: Capitalize TIROS.

corrected

13) P. 13 a) Line 14: insert "near the surface" between "high sensitivity" and "to window channels"

corrected

b) Line 15: Here and elsewhere: Use decimal points to separate the unit and tenth decimal places – at first I was completely confused when I saw 1.080 to 1.150 cm-1, thinking these were near-IR to microwave channels when talking about an IR instrument. Please rewrite simply as "1080 to 1150 cm-1")

corrected

14) P. 14 a) Last line: What does "ARPEGE" stand for?

It is the global NWP model at Météo-France as describe in Section 4.2.2.

15) P. 15 a) Last line: It should also be noted that there is a temporal difference in the overpass times between IASI and CrIS. Also, the Author should always be explicit as to what satellites they're talking about. For IASI is it Metop-A, -B or –C? For CrIS is it SuomiNPP or NOAA-20?

As describe in Section 4.2.1, these are the MetOp-A and B satellites for the IASI instrument and Suomi-NPP for the CrIS instrument. In the case of Figure 8, the IASI spectrum was measured from MetOp-B. Information added in the text.

16) P. 16 a) Equation 1: These equations could be expressed better. Why "EXP"? Is it possible to simplify the subscripts by abbreviating them to a simply letter (e.g., replace "simul" with "s", "REF" with "r" or "b" for background, "AMULSE" with "o" for "obs", etc.) Also, why are three variables in boldface?

Indeed, I have simplified the equation for a better clarity.

17) P. 17 a) Figure 9 caption: What does ",rp" mean?

"rp" is the abbreviation for respectively, which I replace here by "resp.".

b) Lines 5-6: Please explain the relevance to the current paper.

As explained above, the CO2-sensitive channels are used to retrieve the temperature. Thus, if these channels have a large first-guess departures, this can have a negative impact on the data assimilation process and temperature retrieval due to a degraded CO2 prior profiles.

18) P. 18 a) Lines 8-10: It is not clear what relevance this has. Are simulations usually "validated" in this manner? Usually what is done is that radiative transfer model (RTM) "obs minus calc" are subjected to empirical bias correction "tuning" using observations such as this (where the RTM is then later used in retrieval algorithms), but that's not "validation."

We agree this is not validation as the number of cases is far too low. In this example, the comparison illustrates that the impact of switching from static CO2, CH4 and O3 profiles to measured ones on simulations in rather small. This case study highlights the potential benefit of this kind of measurements if they were done on a more regular basis and for a wider network of measurement sites. The text has been modified accordingly.

19) P. 19: nothing

20) P. 20: a) Line 14: include "AIRS, CrIS" along with IASI; Combine sentence fragments by replacing "." with "," before "which".

corrected

b) Line 17: insert "level" after "pixel" and insert "or from cloud-cleared spectra as done by the AIRS and NUCAPS systems (Susskind et al., 2003; Smith and Barnet, 2019)." (References provided below)

corrected

c) Line 20: Replace "ozone analyses" with "ozone retrievals"

corrected

d) Line 23: Insert citation "(Nalli et al. 2018)" for the ozone retrieval validation connection.

corrected

e) Line 26: What does "restitutions" mean?

That's a mistake, that's the French word for retrievals.

21) P. 21 a) Line 2: What radiosondes?

radiosondes means in situ soundings made during the APOGEE campaign. I replaced all the "radiosondes" with "in situ soundings".

22) P. 22: The Authors should provide a bit more information on their 1-D VAR retrieval algorithm and then should relate their results using this algorithm with results from established operational algorithms such as NUCAPS (as in the Nalli et al. Reference) and/or AIRS.

I described this algorithm in more details and compared it to what is done at other centres.

23) P. 23 a) Figure 12: Because the Authors aren't even showing AMULSE measurements in these figures, it's important that they related it to their AMULSE work and also tie it back to other work on the subject of ozone validation using ozonesondes.

I have added references related to our work.

b) Figure 12d: Fix typo "Radiosondge" should be "Radiosonde"

Corrected and remplaced by nothing.

24) P. 24 a) Figure 13 Caption: Simplify by rewriting as "As Figure 12, except for IASI. . ."

Corrected

b) How is "validating the quality of our simulations essential for NWP models"?

added in the article.

Âń Indeed, as mentioned above, the a priori information of gases provided to RTTOV is invariant in time and space. In reality these gases such as CO2, CH4, O3 or CO show significant variability, both temporarily and spatially in the atmosphere. In addition, carbon dioxide sensitive channels are often used to extract information on atmospheric temperature. However, the approximation of using a fixed CO2 for the simulation of infrared satellite observations can have a negative impact on the quality of temperature retrieval as shown by the work of (Engelen et al., 2001) from the AIRS instrument. Finally, data assimilation systems have a bias correction method called VarBC (Auligne et al., 2007). However, this method can correct systematic biases that do not necessarily take into account the specificity of the variability of certain gases. Thus, it is preferable to correct biases at the source through an improvement of simulations in Radiative Transfer Models and will improve the analysis of temperature, humidity and wind in global model assimilations.Âż

25) P. 26 a) Line 13: Please provide full citation. REFERENCES (not all-inclusive –

please feel free to include more if you have them, including specifically references to other in situ measurements such as HIPPO, Atom)

Blunden, J. and D. S. Arndt, Eds., 2019: State of the Climate in 2018. Bull. Amer. Meteor. Soc., 100 (9), Si–S305, doi:10.1175/2019BAMSStateoftheClimate.1.

Chahine et al., 2006: AIRS: Improving weather forecasting and providing new data on greenhouse gases, Bull. Amer. Meteorol. Soc, 87, 911-926.

IPCC, 2014: Climate Change 2014: Synthesis Report. Contribution of Working Groups I, II and III to the Fifth Assessment Report of the Intergovernmental Panel on Climate Change [Core Writing Team, R.K. Pachauri and L.A. Meyer (eds.)]. IPCC, Geneva, Switzerland, 151 pp.

Nalli, N. R., et al., 2018: Validation of atmospheric profile retrievals from the SNPP NOAA-Unique Combined Atmospheric Processing System. Part 2: Ozone, IEEE Trans. Geosci. Remote Sens., 56(1), 598-607, doi:10.1109/TGRS.2017.2762600.

Smith, N. and C.D. Barnet, 2019: Uncertainty characterization and propagation in the community long-term infrared Microwave Product System (CLIMCAPS), Remote Sens., 11, 1227, doi:10.3390/rs11101227.

Susskind, J., C.D. Barnet, and J.M. Blaisdell, 2003: Retrieval of atmospheric and surface parameters from AIRS/AMSU/HSB data in the presence of clouds, IEEE. Trans. Geosci. Remote Sens., 41, 390-409.
* * *
[Figure]

**The development of the Atmospheric Measurements by Ultra-Light Spectrometer (AMULSE) greenhouse gas profiling system and its interest in atmospheric applications**

Lilian Joly[1], Olivier Coopmann[2], Vincent Guidard[2], Thomas Decarpenterie[1], Nicolas Dumelié[1], Julien Cousin[1], Jérémie Burgalat[1], Nicolas Chauvin[1], Grégory Albora[1], Rabih Maamary[1], Zineb Miftah El Khair[1], Diane Tzanos[3], Joël Barrié[3], Éric Moulin[3], Patrick Aressy[3], and Anne Belleudy[3]

[1]GSMA, UMR CNRS 7331, Université de Reims, U.F.R. Sciences Exactes et Naturelles, Reims, France
[2]CNRM, Université de Toulouse, Météo-France, CNRS, Toulouse, France (NWP team)
[3]CNRM, Université de Toulouse, Météo-France, CNRS, Toulouse, France (Instrumentation team)

**Correspondence:** L. Joly, GSMA, UMR CNRS 7331, Université de Reims, U.F.R. Sciences Exactes et Naturelles, Reims, France (lilian.joly@univ-reims.fr); O. Coopmann, Météo-France, CNRM/GMAP/OBS, 42 Avenue Gaspard Coriolis, 31057 Toulouse Cedex, France (olivier.coopmann@umr-cnrm.fr);

**Abstract.**

We report in this paper the development of an embedded ultralight spectrometer (< 3 kg) based on tuneable diode laser absorption spectroscopy (with a sampling rate of 24 Hz) in the mid-infrared spectral region. This instrument is dedicated to in-situ measurements of the vertical profile concentrations of three main greenhouse gases: carbon dioxide ($CO_2$), methane

5   ($CH_4$) and water vapour ($H_2O$) via standard weather and tethered balloons. The plug and play instrument is compact, robust, cost-effective, and autonomous. The instrument also has a low power consumption and is non-intrusive.

It was first calibrated during an *in situ* experiment on an ICOS (Integrated Carbon Observation System) site for several days, then used in a two experiments with several balloon flights up to 30 km altitude in the Reims-France in 2017-2018 in collaboration with Météo-France/CNRM Centre National de Recherches Météorologiques.

10   This paper shows the valuable interest of the data measured by AMULSE (Atmospheric Measurements by UltraLight SpEctrometer) instrument during the APOGEE (Atmospheric Profiles Of GreenhousE gasEs) measurement experiment, specifically for the vertical profiles of $CO_2$ and $CH_4$, which remain very sparse. We have carried out several experiments showing that the measured profiles have several applications: for the validation of simulations of infrared satellite observations, for evaluating the quality of chemical profiles from Chemistry Transport Models (CTM) and for evaluating the quality of retrieved chemical

15   profiles from the assimilation of infrared satellite observations. The results show that the simulations of infrared satellite observations from IASI (Infrared Atmospheric Sounding Interferometer) and CrIS (Cross-Track Infrared Sounder) instruments performed in operational mode for Numerical Weather Prediction (NWP) by the Radiative Transfer Model (RTM) RTTOV (Radiative Transfer for TIROS Operational Vertical sounder) are of good quality. We also show that the MOCAGE (MOdèle de Chimie Atmosphérique à Grande Échelle) and CAMS (Copernicus Atmospheric Monitoring Service) CTMs modeled ozone

20   profiles fairly accurately and that the CAMS CTM represents the methane in the troposphere well compared to MOCAGE. Fi-

**Fig. 1.**

[Figure]

---

## Author Comment (AC2) · 5 Feb 2020

Review#2

The evolution of AMULSE (Atmospheric Measurements by Ultra-Light Spectrometer) and its interest in atmospheric applications. Results of the Atmospheric Profiles Of

[Figure]

GreenhousE gasEs (APOGEE) weather balloon release campaign for satellite retrieval validation

Anonymous Referee #2

Thank you very much for your interest in our work. I warmly thank the Reviewer for his valuable comments which enabled us to improve this article.

General Comments

1) This paper should be reconsidered after major revisions.

2) There are sections of the paper that focus solely on IASI analysis and then shoehorn in the fact that the analysis was also done with CrIS. It would be beneficial to add more details on the analysis done with CrIS. A couple of the sections that need more information are mentioned in the specific comments.

3) There are many grammatical mistakes throughout that need fixing. I compiled a list of the ones I found but there are certainly more.

4) Overall, the structure of the paper is good.

Specific comments

1) Many acronyms are used without being spelled out in the abstract (CNRM, AMULSE, APOGEE, IASI, CrIS, NWP, RRTOVRTTOV, MOCAGE, CAMS)

Added

2) P8. Figure 2: What does the % of $H_2O$ represent?

We changed the unit to be more explicit, we put it in ppmv (parts per million by volume).

3) P11. Line 3: The photo synthesis phenomenon enriching $CO_2$ should have a reference
3) In section 5.1.3 and 5.2.2, the authors should add more information on CrIS. CrIS seems to be mentioned off handwhile all of the analysis is done with IASI.

4) In equation 1 in section 5.1.5 exp should be rewritten as either exp(O3) or O3=ln(.....). The superscript simul could also be shortened to just sim.

EXP means Âń experiment Âż for example, EXPO3 means experiment for ozone case, etc.

Technical corrections

P1. Line 5-6: I recommend splitting this into two sentences. Maybe "The plug and play instrument is compact, robust, cost-effective, and autonomous. The instrument also has a low power consumption and is non-intrusive."

Corrected

P2. Line 7-8: rewrite to "The climate of the earth is currently changing quickly. Recently, evidence has accumulated showing that this climate change is directly related to human activities." Line 9: change "involve" to "create" Line 10: change "of" to "on"Line 10: remove "on Earth" Line 20: remove "the" before CH4

All Corrected

Line 22: change "many informations" to "information" Line 25: Define the RRTOV RT-TOV acronym

Corrected

P4. Line 26: change "meters'" to "meters" Line 27: change "lots of preparations" to "a lot of preparation"

Corrected

P5. Line 2: change "This specificity of the balloons, to be able to access the pofiles," to"The balloons' ability to access the profiles"

Corrected

P6. Line 9: close the parentheses Line10: change "The spectrometer weights is lower than" to "The spectrometer weighs lessthan" Line 13: change "send various datas" to "sends various data" Line 14: computerfixed 400m in relation to what? Line 15: capitalize AMULSE Line 23: change "at" to "a"change "about" to "for" Line 25: capitalize AMULSE

Corrected

P7. Figure 1: capitalize AMULSE,remove "of (a) CO2,(b) CH4 and (c) H2O" P9. Line 3: replace "the" with "our" Line 9:change "balloon" to "balloons" Line 10: add period at the need of this sentence Line 18:change "Balloon" to "The balloon" Line 20: change "Balloon" to "The balloon" Line 20: change "cutted" to "cut" Line 29: make hypothesis plural

P11. Move the table 2 captiondown a line Line 6-7: change "prospects for 2020 is to make water vapour measure-ments by laser diode spectrometry to have a better accuracy" to "goals for 2020 is tomake water vapour measurements by laser diode spectrometry more accurate"

Corrected

P12.Line 6: change "a prior" to "a priori"

Corrected P13. Line 15,19,20, and 21: Either use commas or no separation between the thousand and hundred place in numbers instead of decimal points. (i.e. 1,000 or 1000)

Corrected

P16. Need to fix decimal points on this page as well. Line 4 and 5: what does "rp" mean Why did CO2 and CH4 become italicized midway throughthis page

decimal point and italicized corrected. "rp" is the abbreviation for respectively, which I

replace here by "resp.".

P17. Figure 9: If ",rp" stands for reference profiles you need a better way of conveying this Line 5: add "to" between "CO2" and "have"

In the same way Âń rp Âż means Âń respectively Âż

P18. Line 8: change "validate to "verify" and remove "essentially"

Corrected

P19. Line 5: Remove"of the"

Corrected

P20. Line 13: add space to "be retrieved" Line 14: change ". Which" to ", which" Line 25: Remove "of" add the start of the sentence. I do not think "restitutions" is the right word here.

Corrected

P22. Last 2 lines: Remove italics on chemical names

Corrected

P23. Figure 12panel d: change "radiosondage" to "radiosonde"

Corrected
* * *
[Figure]

**The development of the Atmospheric Measurements by Ultra-Light Spectrometer (AMULSE) greenhouse gas profiling system and its interest in atmospheric applications**

Lilian Joly[1], Olivier Coopmann[2], Vincent Guidard[2], Thomas Decarpenterie[1], Nicolas Dumelié[1], Julien Cousin[1], Jéremie Burgalat[1], Nicolas Chauvin[1], Grégory Albora[1], Rabih Maamary[1], Zineb Miftah El Khair[1], Diane Tzanos[3], Joël Barrié[3], Éric Moulin[3], Patrick Aressy[3], and Anne Belleudy[3]

[1]GSMA, UMR CNRS 7331, Université de Reims, U.F.R. Sciences Exactes et Naturelles, Reims, France
[2]CNRM, Université de Toulouse, Météo-France, CNRS, Toulouse, France (NWP team)
[3]CNRM, Université de Toulouse, Météo-France, CNRS, Toulouse, France (Instrumentation team)

**Correspondence:** L. Joly, GSMA, UMR CNRS 7331, Université de Reims, U.F.R. Sciences Exactes et Naturelles, Reims, France (lilian.joly@univ-reims.fr); O. Coopmann, Météo-France, CNRM/GMAP/OBS, 42 Avenue Gaspard Coriolis, 31057 Toulouse Cedex, France (olivier.coopmann@umr-cnrm.fr);

**Abstract.**

We report in this paper the development of an embedded ultralight spectrometer (< 3 kg) based on tuneable diode laser absorption spectroscopy (with a sampling rate of 24 Hz) in the mid-infrared spectral region. This instrument is dedicated to in-situ measurements of the vertical profile concentrations of three main greenhouse gases: carbon dioxide ($CO_2$), methane

5   ($CH_4$) and water vapour ($H_2O$) via standard weather and tethered balloons. The plug and play instrument is compact, robust, cost-effective, and autonomous. The instrument also has a low power consumption and is non-intrusive.

It was first calibrated during an *in situ* experiment on an ICOS (Integrated Carbon Observation System) site for several days, then used in a two experiments with several balloon flights up to 30 km altitude in the Reims-France in 2017-2018 in collaboration with Météo-France/CNRM Centre National de Recherches Météorologiques.

10   This paper shows the valuable interest of the data measured by AMULSE (Atmospheric Measurements by UltraLight SpEctrometer) instrument during the APOGEE (Atmospheric Profiles Of GreenhousE gasEs) measurement experiment, specifically for the vertical profiles of $CO_2$ and $CH_4$, which remain very sparse. We have carried out several experiments showing that the measured profiles have several applications: for the validation of simulations of infrared satellite observations, for evaluating the quality of chemical profiles from Chemistry Transport Models (CTM) and for evaluating the quality of retrieved chemical

15   profiles from the assimilation of infrared satellite observations. The results show that the simulations of infrared satellite observations from IASI (Infrared Atmospheric Sounding Interferometer) and CrIS (Cross-Track Infrared Sounder) instruments performed in operational mode for Numerical Weather Prediction (NWP) by the Radiative Transfer Model (RTM) RTTOV (Radiative Transfer for TIROS Operational Vertical sounder) are of good quality. We also show that the MOCAGE (MOdèle de Chimie Atmosphérique à Grande Échelle) and CAMS (Copernicus Atmospheric Monitoring Service) CTMs modeled ozone

20   profiles fairly accurately and that the CAMS CTM represents the methane in the troposphere well compared to MOCAGE. Fi-

**Fig. 1.**

---

## Author Response (AR2)

Thank you very much for helping us to increase the quality of this article. Here below are the answers to your remarks.

A few specific comments:

P2,L23: "thermodynamic structure and atmospheric composition"
Done

P6L5-7: "DFB" = distributed feedback; "ICL" = interband cascade laser
Done

P6,L13: "Judson" is "Teledyne Judson Technologies"; PTU should be explained
Done

P6,L14: "PTU" = pressure, temperature and humidity
Done

P6,L23: "sensitivity" represents a signal/analyte quantity--do you mean "detection limit" (over what time?)? Table 1 shows values for both CH4 and CO2 in 2015 while P5, L21-22 states that in 2015 AMULSE was a single gas CH4 methane instrument and the dual-gas version was introduced in 2016. Table captions go at the top of the table, figure captions below.

I mean sensitivity. Indeed, the dual-gas version was introduced in 2016. Before that we had two instruments, a CO2 single gas version and a CH4 single gas version. I added an explanation (with two single-gas versions CO2 and CH4) in the caption and put it at the top.

Figure 1: comment in text on CO2 drift from early to late part of time series. Average of the difference does not capture--average absolute difference would be a better metric in this case. Stating the std dev as you have is valuable. Greater color difference between traces would be helpful.

This is the absolute difference I use, I'll notify it in the text and I changed the color of the curves (changed in main text).

P11,L7: "CO2 near the surface", "at night"; one reviewer suggested a reference for this. The artifact in H2O due to outgassing on ascent is mentioned, which explains the non-physical values shown in Fig. 5c, but why aren't descent data shown--the text mentions valid data then. A temperature profile plot would be a useful addition to Fig 5 and help with identification of the tropopause.

I added a publication for the day-night cycle of atmospheric CO2 concentration (due to photosynthesis) \citep{Schmidt2014} add in bibtex. I also added the temperature to better identify the tropopause level. I also changed the description of the figure.

P15,L9: interpolate or average? Doesn't the in situ data have higher vertical resolution so that you would need to average the data over the depth of a model layer? Fig 7-- would probably be better to put the reference trace on top, so that it is clearer that the profile in the shaded (non-measured) region is not the from the sonde.

That is indeed what was done in this case. The in-situ profiles were interpolated to the fixed pressure levels by averaging the layers centred on these levels.

Modification in the text:

«… ,we have interpolated the in-situ profiles on the 101 fixed pressure levels by making layer averages centred on these levels. »

This is an interesting idea for Figure 7. It has been modified accordingly.

P16,L11: "red"? Red might actually be better

Also good idea, Figure 8 has been modified.

P16,L13: "This" here is confusing since it naturally refers to the "good agreement" when you are addressing the observed differences in the window regions. "The difference seen between the simulated IASI and CrIS window channels is due to the temporal and spatial differences between the selected pixels."

Corrected in the document with this suggestion.

« The difference seen between the simulated IASI and CrIS window channels is due to the temporal and spatial differences between the selected pixels, which implies a different skin temperature used in RTTOV »

P17,L7+: as a reviewer asked/suggested, explain 'EXP'. You did so in your response, but not in the manuscript.

Added in the paper :

« Figure 5 shows the difference between simulated brightness temperature ($BT^S$) with $O_3$ in purple (resp. $CO_2$ in red or $CH_4$ in orange) *in situ* measurement profiles ($X^M$) and simulated brightness temperature with $O_3$ (resp. $CO_2$ or $CH_4$) reference profiles ($X^R$) for *in situ* sounding from 2018-04-17 at 10 UTC for all channels contained in the IASI (a) and CrIS (b) spectrum, where EXP means experiment (for example, EXPO3 means experiment for ozone case, etc.), such as: »

P18: adjust colors in Fig. 9 to be more distinct. Clearer structure in the figure caption. Why is SONDE capitalized?

Figure 9 has been reconstructed.
SONDE is in capital to mark the comparison to AMULSE

P18,L8: What "specific case"? "For the case of 2018-04-07, it can be seen in Fig 9 that..."

Changed in the text:

« For the case of 2018-04-07, it can be seen in Figure 9 that our static reference profiles of carbon dioxide and methane lead to realistic simulations, as for *in situ* profiles »

P20,L24: reference for the higher difficulty in CH4 simulation in CTMs (just source term or chemistry as well?) and the major scientific questions regarding CH4 trends.

Specific references to these issues have been added in this paper.

Archibald et al., 2020 « Description and evaluation of the UKCA stratosphere--troposphere chemistry scheme (StratTrop vn 1.0) implemented in UKESM1 »

Houweling et al., 2017 « Global inverse modeling of CH4 sources and sinks: an overview of methods »

Naik et al., 2013 « Preindustrial to present-day changes in tropospheric hydroxyl radical and methane lifetime from the Atmospheric Chemistry and Climate Model Intercomparison Project (ACCMIP) »

Non-public comments to the Author:

There are some usage and construction issues that make the manuscript a little difficult to parse and understand in places. I believe that much of this will be addressed during the copyediting state, and I encourage you to work with the Copernicus copyediting professionals at that time. One high profile example is the usage "its interest in" in the title and elsewhere, which implies that AMULSE is interested in atmospheric applications. A better construction might be more simply "and application for satellite retrieval validation".

I agree, I changed the title.